# Stochastic Encodings for Active Feature Acquisition

**Alexander Norcliffe**[1]   **Changhee Lee**[2]   **Fergus Imrie**[3]   **Mihaela van der Schaar**[4]   **Pietro Liò**[1]

## Abstract

Active Feature Acquisition is an instance-wise, sequential decision making problem. The aim is to dynamically select which feature to measure based on current observations, independently for each test instance. Common approaches either use Reinforcement Learning, which experiences training difficulties, or greedily maximize the conditional mutual information of the label and unobserved features, which makes myopic acquisitions. To address these shortcomings, we introduce a latent variable model, trained in a supervised manner. Acquisitions are made by reasoning about the features across many possible unobserved realizations in a stochastic latent space. Extensive evaluation on a large range of synthetic and real datasets demonstrates that our approach reliably outperforms a diverse set of baselines.

## 1. Introduction

The standard supervised learning paradigm is to learn a predictive model using a training dataset of features and labels, such that the model can make accurate predictions on unseen test inputs. A fundamental assumption is that, at test time, all features are jointly available. However, this assumption does not always hold. Consider the example of a doctor diagnosing a patient (Kachuee et al., 2019b;a). Initially, there is little to no information available and, while there are many tests that could be conducted, the doctor will choose which ones to carry out based on their current understanding of the specific patient's condition. For instance, if a patient has pain in their leg, and the doctor suspects a fracture, a leg X-ray might be prioritized. Active Feature Acquisition (AFA)[1] is an inference time task, where the features are not assumed to be all available at once. Instead, on an instance-wise basis, a model sequentially acquires features based on the existing observations to best aid long-term prediction. A common approach is to use Reinforcement Learning (RL) (Rückstieß et al., 2013; Shim et al., 2018), since this is a natural solution to a sequential decision making problem. However, RL suffers from training difficulties such as sparse reward, exploration vs exploitation, and the deadly-triad (Henderson et al., 2018; Erion et al., 2022; Van Hasselt et al., 2018). An alternative approach is to select features that *greedily* maximize the conditional mutual information (CMI) (Chen et al., 2015a;b). This has a significant drawback: CMI does not capture the effects of unobserved features that can be acquired at a later stage. This results in myopic decision making that favors immediate predictive power over sets of jointly informative features, or features that are highly informative of which feature to acquire next. Additionally, we argue that CMI is not even guaranteed to be the best short-term objective from the perspective of the 0-1 Loss (making a single decisive prediction). Since maximizing CMI is equivalent to minimizing entropy, and this can be achieved by making unlikely classes even more unlikely, rather than distinguishing between more probable outcomes. We further explore the drawbacks of CMI in Section 4.

Motivated by the shortcomings of RL and CMI maximization, we introduce a novel AFA approach, which we call **S**tochastic **E**ncodings for **F**eature **A**cquisition (**SEFA**), that departs from existing methods in several key ways. First, we shift the acquisition problem from reasoning in a complex feature space to a latent space. We use an information bottleneck-inspired term (Tishby et al., 1999) to regularize the latent space such that decisions are made using label-relevant information only and not noise associated with the features. Second, we use *stochastic* encoders, allowing us to acquire features by considering their effect across a diverse range of possible latent realizations. This allows us to capture the effects of features that are yet to be observed, resulting in acquisitions that are non-greedy by design. Third, our acquisition objective places more focus on labels with higher predicted likelihood, leading to acquisitions that help

---

[1]Department of Computer Science and Technology, University of Cambridge, Cambridge, United Kingdom [2]Department of Artificial Intelligence, Korea University, Seoul, Korea [3]Department of Statistics, University of Oxford, Oxford, United Kingdom [4]Department of Applied Mathematics and Theoretical Physics, University of Cambridge, Cambridge, United Kingdom. Correspondence to: Alexander Norcliffe <alin2@cam.ac.uk>.

*Proceedings of the 42nd International Conference on Machine Learning*, Vancouver, Canada. PMLR 267, 2025. Copyright 2025 by the author(s).

---

[1]This problem has also been referred to as Dynamic Feature Selection (DFS).

to disambiguate between the most likely classes. Finally, to avoid the difficulties posed by RL, we do not train our model to make acquisitions directly. Instead, we train with a predictive loss and make acquisitions by maximizing a custom objective in a suitably regularized latent space. Our contributions are as follows: (1) We re-examine the CMI acquisition objective and provide theoretical reasoning and concrete examples of its sub-optimality. (2) We introduce SEFA, our novel AFA approach motivated by the limitations of RL and CMI maximization. (3) We evaluate SEFA on multiple synthetic and real-world datasets, including cancer classification tasks. Comparing against various state-of-the-art AFA baselines, we see that SEFA consistently outperforms these methods. Extensive ablations further demonstrate each novel design choice is required for the best performance.

## 2. Related Work

**Reinforcement Learning.** The most common AFA approach is to frame the problem as a Markov Decision Process and train a policy network with RL to decide which feature to acquire next (Dulac-Arnold et al., 2011; Rückstieß et al., 2013; Shim et al., 2018; Janisch et al., 2019; Mnih et al., 2014; Kachuee et al., 2019a). The RL approach readily extends to a temporal setting where features and labels can change over time (Kossen et al., 2023; Yin et al., 2020). Whilst a natural solution to AFA, RL suffers from training difficulties, and various advances in the RL field have been applied to account for this. For example, using generative models to augment datasets (Zannone et al., 2019), providing mutual information as additional input to the policy (Li & Oliva, 2021), using gradient information in the training process (Ghosh & Lan, 2023), and reward shaping (Peng et al., 2018).

**Conditional Mutual Information Maximization.** Conditional mutual information tells us how much we can learn about one variable by measuring a second, whilst already knowing a third. Greedy CMI maximization is a common AFA approach, due to its grounding in information theory. However, as we demonstrate in Section 4, it inherently makes short-term acquisitions and is prone to making acquisitions that do not distinguish between likely labels. Among existing approaches, networks can be trained to directly predict CMI (Gadgil et al., 2024), or policy networks can be specially trained to maximize CMI without ever calculating it (Chattopadhyay et al., 2023; Covert et al., 2023). Generative models are a second way to estimate CMI by taking Monte Carlo estimates over conditional distributions, (Ma et al., 2019; Chattopadhyay et al., 2022; Rangrej & Clark, 2021; Early et al., 2016). This approach suffers from associated generative modeling challenges, producing poor estimates of CMI, thus adding to the limitations. Improved performance can be achieved with advances in generative

modeling (Peis et al., 2022; He et al., 2022; Li et al., 2020; Li & Oliva, 2020).

**Alternative Solutions.** Sensitivity-based solutions make selections based on how sensitive the label is to a given feature (Kachuee et al., 2017; 2018). However, since missing values are filled with zero and measuring a feature is discontinuous, the gradient does not reliably represent the true sensitivity. Imitation learning has been applied (Valancius et al., 2024; He et al., 2016), however, this requires access to an oracle or to construct one. Prior to deep learning, decision trees were used, with features acquired at each branch of a tree if the feature is unobserved (Xu et al., 2012; 2013; Kusner et al., 2014; Trapeznikov & Saligrama, 2013; Xu et al., 2014). This has also been generalized to ensembles of decision trees (Nan et al., 2015; 2016).

## 3. Active Feature Acquisition

**Problem Setup.** In standard $C$-way classification, we have a $d$-dimensional feature vector given by the random variable $X \in \mathcal{X}$ with realization $\mathbf{x} = (x_1, x_2, \ldots, x_d)$, and a label given by $Y \in [C]$ with realization $y$. Ordinarily, we assume all features are observed; however, more generally, we wish to allow arbitrary feature subsets as valid inputs. Therefore, let $*$ represent a missing feature value and $\mathcal{X} = \prod_{i=1}^{d}(\mathcal{X}_i \cup \{*\})$. We denote an input with feature subset $S \subseteq [d]$, as $\mathbf{x}_S$, where $x_{S,i} = x_i$ if $i \in S$, and $x_{S,i} = *$ if $i \notin S$. Given a training set, $\mathcal{D}_{\text{Train}} = \{(\mathbf{x}_S, y)_n\}_{n=1}^{N}$, the AFA task is to train a model that takes a *test* instance with arbitrary observations $\mathbf{x}_O$, and iteratively acquires new features. The model's long-term acquisition goal is to acquire a sequence of features $S^*$ to maximize the confidence in its prediction whilst minimizing the number of acquired features:

$$S^* = \underset{S \in [d] \setminus O}{\arg\max} \left( \max_{c \in [C]} p_{\text{Model}}(Y = c | \mathbf{x}_{O \cup S}) - \lambda |S| \right)$$
$$\text{subject to} \quad |S| \leq B.$$

Here, $\lambda$ balances how much we optimize for a confident prediction compared to limiting the number of acquisitions, and $B$ is a given feature budget. These parameters are highly domain-dependent. For example, in medicine, where the stakes are high, we have large $B$ and low $\lambda$. There is a high tolerance for acquiring features if we make confident predictions. In this paper, we assume we are able to acquire all features if necessary and, unless otherwise stated, models are evaluated by tracking their predictive performance during an entire acquisition starting with no features.

**Acquisition in Practice.** The standard approach to AFA is to construct an acquisition objective function $R : \mathcal{X} \times [d] \to \mathbb{R}_{\geq 0}$, that scores each feature conditioned on existing observations, and to acquire the feature that maximizes this: $i^* = \arg\max_{i \in [d] \setminus O} R(\mathbf{x}_O, i)$. The objective is defined

by the method. For instance, CMI methods use the CMI: $R_{\text{CMI}}(\mathbf{x}_O, i) = I(X_i; Y|\mathbf{x}_O)$, telling us how much measuring $X_i$ will reduce the entropy of $Y$ conditioned on $\mathbf{x}_O$. RL methods use the output of a policy or Q network, trained directly on the sequential feature acquisition problem: $R_{\text{RL}}(\mathbf{x}_O, i) = Q_\theta(\mathbf{x}_O)_i$. Following an acquisition, we update the observed feature set to be $O \cup i^*$ and repeat the acquisition process.

## 4. Limitations of CMI Maximization

Here we more closely examine the shortcomings of greedy CMI maximization for AFA, to gain an understanding of *why* CMI maximization can be sub-optimal and how this can be addressed. Whilst grounded in theory and extensively applied, it suffers from two drawbacks.

First, greedy CMI maximization makes myopic acquisitions, which in some scenarios is *guaranteed* to be sub-optimal. We prove this with an example. Consider a feature vector with $d + 1$ features, the first $d$ of which are binary, and the last taking an integer value from 1 to $d$: $X \in \{0, 1\}^d \times [d]$. The final feature acts as an indicator, informing us which of the other $d$ features gives the label, $y = x_{x_{d+1}}$. The optimal strategy is to first choose the indicator then its designated feature, requiring two acquisitions. However, the expected number of acquisitions to arrive at the *same* prediction by greedily maximizing CMI is $3 - \frac{1}{d}$ (proof in Appendix F). The theoretical insight into why CMI fails is because possible future observations are not considered in the present decision since they are marginalized out, $p(x_i, y|\mathbf{x}_O) = \int p(x_j, x_i, y|\mathbf{x}_O)dx_j$. Each acquisition is made as if there are no subsequent acquisitions and therefore the indicator is not chosen first. This is not specific to CMI, but any scoring that marginalizes out unobserved features.

**Proposition 4.1.** *Any acquisition objective that uses the marginal $p(x_i, y)$ to score feature $i$ will not select the indicator first.*

The proof is straightforward: With no other features, the indicator and label are independent, $p(x_{d+1}, y) = p(x_{d+1})p(y)$. It is therefore impossible to measure its effect on the label without considering possible values of other features, regardless of how the effect is measured. RL methods do not suffer from this, since during training different scenarios are seen and the effects distilled into the parameters. Building on this, an adjusted objective that uses CMI but also considers possible unobserved feature values can solve the indicator problem under greedy maximization.

**Proposition 4.2.** *Greedy maximization of $\mathbb{E}_{p(\mathbf{x}_U|\mathbf{x}_O)} I(X_i; Y|\mathbf{x}_O, \mathbf{x}_U)$ is an optimal strategy for the indicator problem, where $U = [d] \setminus (O \cup i)$.*

We prove this in Appendix F. Note we do *not* use this as our acquisition objective, since this is intractable. The key

takeaway from these two propositions is that considering possible values of other unobserved features is *necessary* for optimality and, if the objective is chosen well, *sufficient*. Whilst we used the indicator example to show this, more generally it applies to settings with features that are jointly informative but individually uninformative - CMI does not acquire jointly informative features because it marginalizes out their interdependencies with the label.

The second drawback of CMI is that, even as a short-term objective, it is not guaranteed to be the best objective for identifying the single most likely class. CMI maximization is equivalent to minimizing entropy, and to show why this is not guaranteed to be optimal, consider two distributions over three classes: $H([0.5, 0.5, 0.0]) = 0.693$ and $H([0.7, 0.15, 0.15]) = 0.819$. The first distribution has lower entropy, but the second identifies the most likely class. The insight is that it is possible to maximize CMI by reducing some class probabilities to be as low as possible. However, we often desire to identify the most likely class, rather than ruling out options that may have already had a low probability. Therefore, an effective acquisition objective will place focus on distinguishing between the most likely labels. We provide another example in Appendix G.

As an aside, from the perspective of Rényi entropies (Rényi, 1961), CMI maximization minimizes the Shannon entropy,[2] $H_1(p) = -\sum_i p_i \log(p_i)$. Whereas if the aim is to identify the single most likely class, minimizing the min-entropy $H_\infty(p) = -\log \max_i p_i$, might be more appropriate. However, this would still suffer from making myopic acquisitions. More importantly, the theoretical foundation no longer holds: minimizing Shannon entropy *is* equivalent to maximizing the KL divergence; however, minimizing $H_\infty$ is *not* equivalent to maximizing the $D_\infty$ divergence.

## 5. Method

To address the limitations of RL and CMI for AFA, we propose a novel method, called **S**tochastic **E**ncodings for **F**eature **A**cquisition (**SEFA**). We provide a block diagram of our method in Figure 1, showing both how the model makes predictions and calculates the acquisition objective. We describe the architecture and training in Section 5.1 and the acquisition objective in Section 5.2.

### 5.1. Architecture and Training

SEFA uses an encoder-predictor architecture with intermediate stochastic latent variable $Z \in \mathcal{Z}$. Predictions are given by $p_{\theta,\phi}(y|\mathbf{x}_S) = \mathbb{E}_{p_\theta(\mathbf{z}|\mathbf{x}_S)} p_\phi(y|\mathbf{z})$.

**Encoder.** In order to use neural networks to implement

---

[2]Throughout the paper, unless specified, entropy refers to Shannon entropy.

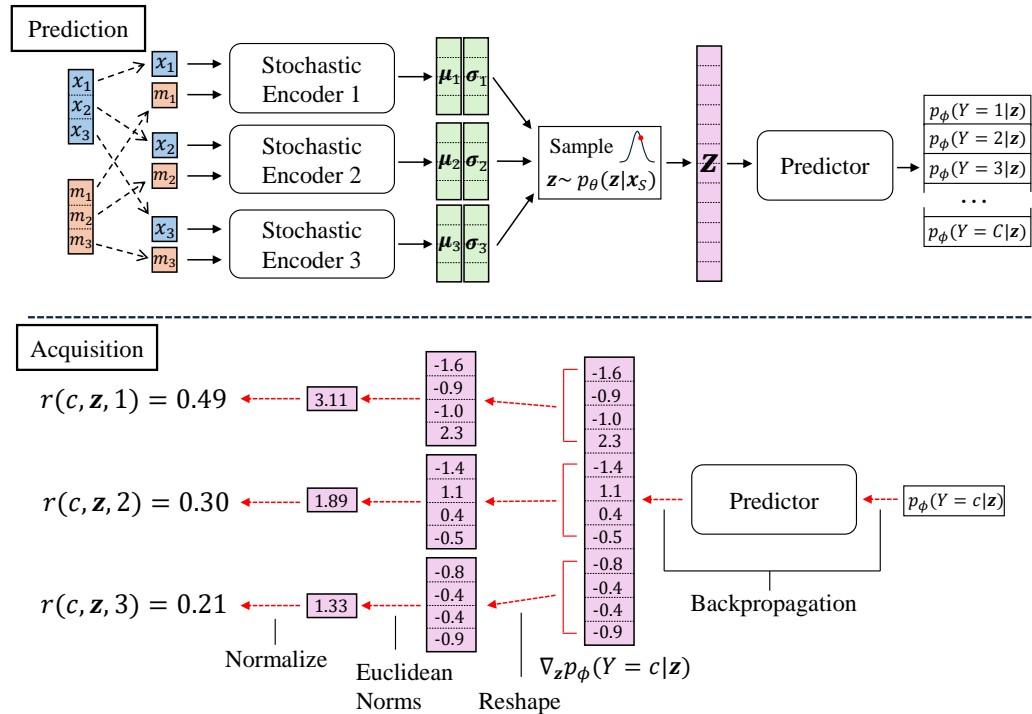

Figure 1. Block diagram of SEFA. Illustrated using three features and four latent components per feature. The presence or absence of a feature value is indicated with a binary mask vector $\mathbf{m}$. Prediction and acquisition scoring with one latent sample is given together with example numerical values for acquisition.

SEFA, the input vector must not change size. Therefore, to account for missing values, we impute missing feature values and use a binary mask, $M \in \{0,1\}^d$, as additional input to the encoder to signal if a value is real or imputed. $x_{S,i} = x_i$, $m_{S,i} = 1$ if $i \in S$ and $x_{S,i} = *$, $m_{S,i} = 0$ if $i \notin S$. Where $*$ is 0 for continuous features and a new category for categorical features.

For reasons that become clear when we describe the acquisition objective (see Section 5.2), we factorize the latent space, so that each feature is separately encoded to $l$ latent components (see Figure 1)

$$p_\theta(\mathbf{z}|\mathbf{x}_S) = \prod_{i=1}^{d} p_{\theta_i}(\mathbf{z}_{\mathcal{G}_i}|x_{S,i}, m_{S,i}).$$

Here $\mathcal{G}_i$ indexes the latent components that feature $i$ is responsible for encoding. For example, if $l = 3$ then $\mathcal{G}_1 = \{1,2,3\}$, $\mathcal{G}_2 = \{4,5,6\}$ etc. Each encoder is an MLP, $f_{\theta_i}^e : \mathcal{X}_i \times \{0,1\} \to \mathbb{R}^l \times \mathbb{R}_{>0}^l$, that outputs a mean and diagonal standard deviation of a normal distribution.

**Predictor.** We make predictions on individual latent samples with a predictor network given by an MLP, $f_\phi^p : \mathbb{R}^{ld} \to \Delta_C$, that predicts a probability distribution over $C$ classes.

**Training.** The SEFA architecture is trained in a supervised manner, minimizing the predictive negative log-likelihood,

avoiding potential issues associated with training using RL. We also impose an information bottleneck regularization term on the latent space $I(X_S; Z)$ (Tishby et al., 1999), minimizing how much information about the features is encoded in the latent variable. The regularization term is implemented using a variational upper bound (Alemi et al., 2017), as the expected KL divergence between $p_\theta(\mathbf{z}|\mathbf{x}_S)$ and a prior $p(\mathbf{z})$. We use $\mathcal{N}(0,1)$ as the prior so that the KL divergence has a closed-form expression. This is multiplied by $\beta$ to balance predictive performance with regularization. Additionally, to train SEFA to make predictions on arbitrary feature subsets, at each training step we first uniformly sample a removal probability, and then each feature has that probability of being removed. This gives the loss

$$L = \mathop{\mathbb{E}}_{p_{\text{Data}}(\mathbf{x}_S, y)} \mathop{\mathbb{E}}_{p_{\text{Subsample}}(S')} \Bigg[ -\log\big(p_{\theta,\phi}(y|\mathbf{x}_{S \cap S'})\big)$$
$$+ \beta D_{\text{KL}}\big(p_\theta(Z|\mathbf{x}_{S \cap S'})||p(Z)\big) \Bigg]. \quad (1)$$

### 5.2. Acquisition

Once the SEFA architecture is trained, we can make acquisitions using our novel acquisition objective

$$R(\mathbf{x}_O, i) = \sum_{c \in [C]} p_{\theta,\phi}(Y = c|\mathbf{x}_O) \mathop{\mathbb{E}}_{p_\theta(\mathbf{z}|\mathbf{x}_O)} r(c, \mathbf{z}, i) \quad (2)$$

where $r : [C] \times \mathcal{Z} \times [d] \to \mathbb{R}_{\geq 0}$ is a scoring function

$$r(c, \mathbf{z}, i) = \frac{||\mathbf{g}_{\mathcal{G}_i}||_2}{\sum_j ||\mathbf{g}_{\mathcal{G}_j}||_2}, \quad \mathbf{g} = \nabla_{\mathbf{z}} p_\phi(Y = c|\mathbf{z}). \quad (3)$$

Recall $\mathcal{G}_i$ indexes the components of the latent gradient that feature $i$ encodes. Below we go through this objective, understanding the purpose of each technical component:

**Scoring Function as an Importance Measure.** For a given latent representation $\mathbf{z}$, and a given class $c$, we can use the gradient $\nabla_{\mathbf{z}} p_\phi(Y = c|\mathbf{z})$, to determine how important each component of $\mathbf{z}$ is for making that prediction. Gradients are an established feature attribution technique as a local explainer (Baehrens et al., 2010; Simonyan et al., 2014; Ribeiro et al., 2016), our novelty is to use them in the latent space, not the feature space.

In order to convert the gradients of $\mathbf{z}$ to feature scores, we calculate the length of the gradient in a feature's associated latent components, $||\mathbf{g}_{\mathcal{G}_i}||_2$. Note that this is only possible because we used feature-wise encoders rather than one fully connected encoder. Finally, we normalize scores to treat each latent sample equally, removing the effect of the total gradient length. See Figure 1 for an example calculation.

**Stochastic Encodings.** $r(c, \mathbf{z}, i)$ only tells us how important feature $i$'s latent components are for predicting class $c$, for a *single* $\mathbf{z}$. However, as demonstrated in Section 4, considering possible values of other unobserved features is necessary for optimality, and lacking in CMI. By using stochastic encoders and taking an expectation of $r(c, \mathbf{z}, i)$ over $p_\theta(\mathbf{z}|\mathbf{x}_O)$, multiple possible latent realizations (including those associated with different unobserved feature values) are taken into account in the current decision. We can *analogously* think of this inner loop as conducting Monte Carlo tree search (Coulom, 2006) through possible future observations and measuring which feature on average is most important for predicting the likelihood of class $c$. To fully sample the possible realizations, multiple latent samples are taken; we empirically verify this and the use of stochastic encoders in Section 6 and Appendix C.

**Probability Weighting.** The inner loop of the objective scores feature $i$, based on its expected importance for predicting the probability of class $c$, over many possible unobserved latent realizations. To aggregate across classes, rather than treating each class equally, we take a weighted sum of the scores, using the current predicted probabilities $p_{\theta,\phi}(Y = c|\mathbf{x}_O)$. This places more focus on the more likely classes, overcoming the issue with CMI maximization being possible by making low probabilities lower (see Section 4). Empirically, we verify this in ablations in Appendix C.

**Supervised Training.** Since our acquisition objective is hand-crafted, we can train with a supervised loss, avoiding training difficulties associated with RL.

### 5.3. Understanding the Latent Space

**Benefits.** The remaining novelty of SEFA is the use of the latent space to calculate the acquisition objective. However, it is possible to calculate the objective entirely in the feature space. Here we would train a predictive model $p_{\tilde{\phi}}(y|\mathbf{x}_S)$ and a generative model $p_{\tilde{\theta}}(\mathbf{x}|\mathbf{x}_S)$, and the objective is

$$\tilde{R}(\mathbf{x}_O, i) = \sum_{c \in [C]} p_{\tilde{\phi}}(Y = c|\mathbf{x}_O) \underset{p_{\tilde{\theta}}(\mathbf{x}|\mathbf{x}_O)}{\mathbb{E}} \tilde{r}(c, \mathbf{x}, i).$$

In this case, each component of the gradient maps directly to one feature, so the scoring function simplifies to

$$\tilde{r}(c, \mathbf{x}, i) = \frac{|g_i|}{\sum_j |g_j|}, \quad \mathbf{g} = \nabla_{\mathbf{x}} p_{\tilde{\phi}}(Y = c|\mathbf{x}).$$

We use the latent space instead of the feature space because it allows us to calculate the acquisition objective using *representations* of the features, rather than the features themselves. The benefits are three-fold: (1) Gradients in the latent space are more meaningful and comparable, since all latent components are continuous, and at a similar scale (if there is appropriate regularization). Whereas features can be categorical or at different inherent scales. (2) The information bottleneck regularization removes feature-level noise. Therefore, in the latent space we calculate the acquisition objective using only label-relevant information, instead of taking gradients with noisy feature values. (3) We do not need to train a generative model, and therefore avoid associated complexities such as continuous and categorical variables, multi-modal densities, and highly correlated features. Our ablations in Section 6 and Appendix C verify the use of the latent space over the feature space for calculating the acquisition objective.

**Independent Latent Components.** One of the modeling choices to make the acquisition objective tractable is to encode each feature separately. However, this means measuring one feature does not affect the latent distribution of another feature, even if they are highly correlated. Naively, this decoupling should harm the performance. However, the complex interdependencies can be accounted for by the predictor network $p_\phi(y|\mathbf{z})$. For example, even though measuring feature 1 does not affect feature 2's latent distribution, the predictor network can learn to treat feature 2's latent samples differently based on the value of feature 1's latent samples (whose distribution *does* change after measurement). Therefore, the gradients change, and measuring feature 1 *does* affect the score for feature 2 (and all other features), despite the latent distribution not changing.

**Revisiting the Loss.** The predictive part of the loss in (1) is given by the negative log-likelihood $-\log \mathbb{E}_{p_\theta(\mathbf{z}|\mathbf{x}_S)} p_\phi(y|\mathbf{z})$. It is standard to take one latent sample to calculate this. However, we take multiple samples for two reasons. (1) The predictor network must see enough

latent samples during training to learn to adapt based on different samples during acquisition. (2) If only one sample is taken for a given $\mathbf{x}_S$, the predictive loss encourages each sample to output the same prediction. Either the encoder $p_\theta(\mathbf{z}|\mathbf{x}_S)$ reduces the latent standard deviation, or the predictor $p_\phi(y|\mathbf{z})$ predicts the same label distribution for the different latent samples. Either way, there is a lack of diversity in the latent space predictions. If multiple samples are taken, this allows individual samples' predictions to vary more. This will not improve the predictions, but improves the acquisition since a more diverse set of possible realizations are used in the inner loop of the objective. We verify this empirically in Section 6 and Appendix C.

# 6. Experiments

Here we evaluate SEFA against various deep AFA baselines. We consider a range of synthetic, tabular, image, and medical datasets. For reproducibility, we provide full experimental details in Appendix K, including hyperparameter choices and training procedures, and full dataset details in Appendix H. The code for our method and experiments is available at https://github.com/a-norcliffe/SEFA.

**Baselines.** We consider six different state-of-the-art baselines: Opportunistic Learning (Kachuee et al., 2019a) and GSMRL (Li & Oliva, 2021) as RL baselines, GDFS (Covert et al., 2023) and DIME (Gadgil et al., 2024) as greedy CMI maximization methods, and ACFlow (Li & Oliva, 2020) and EDDI (Ma et al., 2019) as generative models for CMI maximization. We also use three vanilla baselines: a random ordering of features with an MLP predictor, an MLP with a fixed *global* ordering of features, and a VAE (Kingma, Diederik P and Welling, Max, 2014), which has a separate predictive and generative model to estimate the CMI. Further details about the baselines are given in Appendix I.

## 6.1. Synthetic Datasets

We begin by constructing three synthetic classification tasks (denoted Syn 1-3) based on the synthetic experiments used by Yoon et al. (2019), where we know the optimal instance-wise feature ordering. These are binary classification tasks with eleven features sampled from a standard normal. Three logits are calculated from the first ten features, defined as:

$$\ell_1 = 4x_1x_2, \qquad \ell_2 = \sum_{i=3}^{6} 1.2x_i^2 - 4.2,$$

$$\ell_3 = -10\sin(0.2x_7) + |x_8| + x_9 + e^{-x_{10}} - 2.4$$

The binary label is sampled with $p(Y = 1) = 1/(1 + e^\ell)$. Syn 1 uses $\ell_1$ if $x_{11} < 0$ and $\ell_2$ otherwise. Syn 2 uses $\ell_1$ if $x_{11} < 0$ and $\ell_3$ otherwise. Syn 3 uses $\ell_2$ if $x_{11} < 0$ and $\ell_3$ otherwise. In all cases $x_{11}$ determines which features are important to the prediction, so the optimal strategy is to

acquire $x_{11}$ first and then to acquire the remaining relevant features. Table 1 shows how many features each model acquires until all features relevant to a particular instance are selected (including $x_{11}$). SEFA achieves this in the fewest acquisitions and is close to optimal on all three datasets. Estimating CMI using generative models (ACFlow, EDDI, and VAE) performs worse than the fixed ordering, showing that inaccurate estimation of CMI worsens the issues already associated with its greedy maximization. EDDI, in particular, nearly performs as poorly as random selections and consistently performs poorly across all experiments, since it is only trained to indirectly predict $y$ from $\mathbf{x}_S$ and thus produces inaccurate estimates of CMI and $p(y|\mathbf{x}_S)$.

We investigate which features are acquired by the best four models for Syn 3 (Figure 2). SEFA consistently chooses $x_{11}$ first and then continues to make optimal acquisitions, almost achieving the optimal performance of 5 (Table 1). In contrast, DIME acquires $x_7$ first, since this has the highest mutual information initially, despite not being the best for long-term acquisitions. Therefore, when $x_{11} < 0$, DIME does not start acquiring features 3-6 until acquisition 3. GDFS performs similarly, since it is also trained to maximize CMI. Opportunistic RL tends to make noisy acquisitions, as seen by the red trajectories, demonstrating how it suffers from training difficulties. See Appendix A for equivalent diagrams and analysis for Syn 1 and Syn 2.

**Ablations.** To provide further insight into why SEFA performs well, we conduct ablations on the synthetic datasets (Table 1). We investigate the impact of each of our design choices: removing the latent space regularization ($\beta = 0$); using one latent sample during acquisition; using one latent sample during training; using a deterministic encoder (and therefore no expectation over latent samples); calculating the acquisition objective in the feature space rather than the latent space;[3] and not normalizing the acquisition scores. Note removing probability weighting has no impact on binary classification; we prove this and ablate this component in Appendix C. Removing any of the novel components impacts SEFA's performance. Calculating the acquisition objective in latent space is the most important design choice, followed by using multiple acquisition samples and a stochastic encoder. To better understand the performance differences, in Appendix B we examine acquisition heat maps and carry out sensitivity analyses on $\beta$, number of acquisition samples, number of train samples, and number of latent components per feature.

## 6.2. Datasets with Unknown Feature Orderings

Here, we consider multiple synthetic and real-world datasets where the correct feature ordering is not known a priori.

---

[3]This was achieved by using a VAE for $p_{\tilde{\theta}}(\mathbf{x}|\mathbf{x}_S)$ and an MLP for $p_{\tilde{\phi}}(y|\mathbf{x}_S)$.

*Table 1.* Number of acquisitions to acquire the correct features on the synthetic datasets, the lower the better. We provide the mean and one standard error.

| Model | Syn 1 | Syn 2 | Syn 3 |
|---|---|---|---|
| ACFlow | $7.730 \pm 0.139$ | $7.527 \pm 0.254$ | $9.194 \pm 0.278$ |
| DIME | $4.079 \pm 0.064$ | $4.581 \pm 0.217$ | $5.667 \pm 0.038$ |
| EDDI | $9.197 \pm 0.203$ | $9.214 \pm 0.415$ | $9.794 \pm 0.186$ |
| Fixed MLP | $6.009 \pm 0.000$ | $5.996 \pm 0.000$ | $7.999 \pm 0.000$ |
| GDFS | $4.568 \pm 0.219$ | $4.484 \pm 0.159$ | $5.587 \pm 0.201$ |
| GSMRL | $5.570 \pm 0.127$ | $6.227 \pm 0.185$ | $8.199 \pm 0.067$ |
| Opportunistic RL | $4.201 \pm 0.041$ | $4.846 \pm 0.021$ | $5.850 \pm 0.072$ |
| Random | $9.484 \pm 0.006$ | $9.499 \pm 0.005$ | $9.987 \pm 0.008$ |
| VAE | $6.589 \pm 0.088$ | $6.667 \pm 0.140$ | $7.888 \pm 0.065$ |
| **SEFA (ours)** | $\mathbf{4.017 \pm 0.003}$ | $\mathbf{4.099 \pm 0.009}$ | $\mathbf{5.084 \pm 0.026}$ |

| Ablation | Syn 1 | Syn 2 | Syn 3 |
|---|---|---|---|
| $\beta = 0$ | $4.520 \pm 0.082$ | $4.578 \pm 0.109$ | $5.716 \pm 0.104$ |
| 1 Acquisition Sample | $4.683 \pm 0.030$ | $4.862 \pm 0.035$ | $5.700 \pm 0.027$ |
| 1 Train Sample | $4.421 \pm 0.174$ | $4.713 \pm 0.160$ | $5.188 \pm 0.107$ |
| Deterministic Encoder | $4.593 \pm 0.236$ | $4.773 \pm 0.228$ | $5.744 \pm 0.034$ |
| Feature Space Calculation | $5.111 \pm 0.070$ | $5.461 \pm 0.128$ | $5.977 \pm 0.052$ |
| WO Normalization | $4.036 \pm 0.009$ | $4.104 \pm 0.008$ | $5.101 \pm 0.015$ |
| **SEFA (full)** | $\mathbf{4.017 \pm 0.003}$ | $\mathbf{4.099 \pm 0.009}$ | $\mathbf{5.084 \pm 0.026}$ |

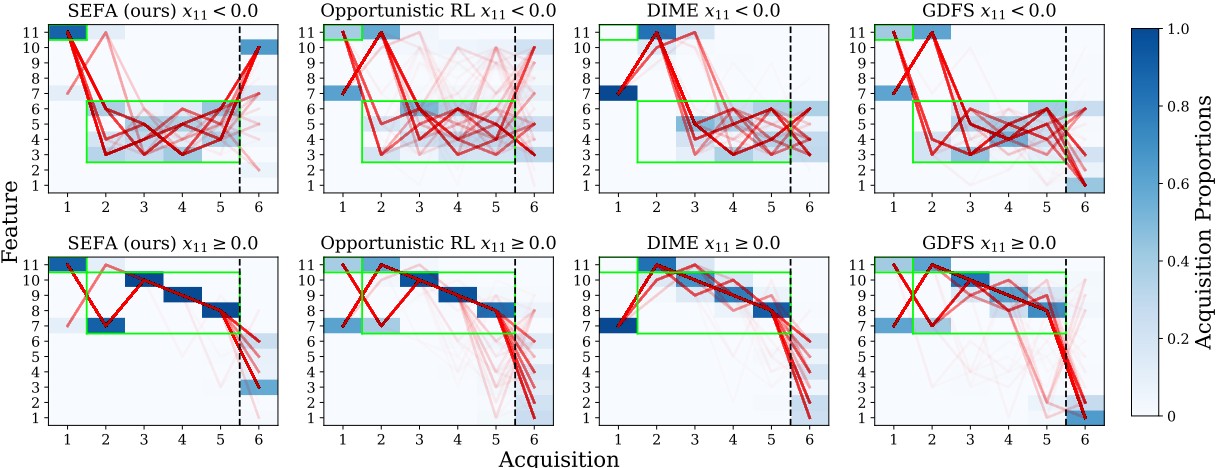

*Figure 2.* Acquisition heat maps and trajectories for Syn 3. Individual trajectories are plotted in red, with the acquisition proportions at each step as a heat map. Green boxes show the optimal strategy, while the vertical black line denotes the minimum number of features required (5).

To evaluate, we start with zero features and calculate the evaluation metric at every step during acquisition. We use AUROC for binary classification tasks and accuracy for multi-class. We report the average metric during acquisition in Table 2 and plot the curves for SEFA, DIME, GDFS, Opportunistic RL, and the fixed MLP ordering in Figure 3, which tend to be the best performing models.

**Cube.** We start with the Cube Synthetic Dataset (Rückstieß et al., 2013; Shim et al., 2018; Zannone et al., 2019). The task is eight-way classification with twenty features. The feature vector is normally distributed around the corners of a cube, with the cube occupying three different dimensions for each class. Irrelevant features are normally distributed around the center. SEFA has the highest average accuracy, and consistently maintains the highest acquisition curve. All

active methods outperform the fixed ordering, except EDDI which suffers from the lack of an inbuilt predictive model.

**Real Tabular.** Next, we consider three real tabular datasets. Bank Marketing (Moro et al., 2014), California Housing (Pace & Barry, 1997) and MiniBooNE (Roe et al., 2005; Roe, 2010). The Bank Marketing dataset is a binary classification task, predicting if a customer subscribes to a product based on marketing data. California Housing consists of features about houses in California districts and the label is the median house price. We converted this into four-way classification by bucketing the labels into four equally sized bins. The MiniBooNE dataset is a particle physics binary classification task to distinguish between electron-neutrinos and muon-neutrinos. In all cases, SEFA has both the highest average evaluation metric and maintains the best evalua-

*Table 2.* Average evaluation metrics during acquisition. Higher values are better; we report the mean and standard error.

| Model | Cube | Bank Marketing | California Housing | MiniBooNE |
|---|---|---|---|---|
| ACFlow | $0.899 \pm 0.001$ | $0.823 \pm 0.013$ | $0.558 \pm 0.010$ | $0.881 \pm 0.018$ |
| DIME | $0.901 \pm 0.001$ | $0.907 \pm 0.002$ | $0.661 \pm 0.002$ | $0.951 \pm 0.001$ |
| EDDI | $0.764 \pm 0.005$ | $0.705 \pm 0.010$ | $0.412 \pm 0.013$ | $0.842 \pm 0.009$ |
| Fixed MLP | $0.883 \pm 0.001$ | $0.909 \pm 0.001$ | $0.658 \pm 0.002$ | $0.954 \pm 0.000$ |
| GDFS | $0.900 \pm 0.000$ | $0.907 \pm 0.001$ | $0.653 \pm 0.002$ | $0.949 \pm 0.000$ |
| GSMRL | $0.891 \pm 0.001$ | $0.879 \pm 0.006$ | $0.638 \pm 0.003$ | $0.946 \pm 0.001$ |
| Opportunistic RL | $0.901 \pm 0.000$ | $0.910 \pm 0.000$ | $0.657 \pm 0.001$ | $0.953 \pm 0.000$ |
| Random | $0.699 \pm 0.001$ | $0.816 \pm 0.003$ | $0.569 \pm 0.003$ | $0.912 \pm 0.001$ |
| VAE | $0.901 \pm 0.001$ | $0.878 \pm 0.002$ | $0.633 \pm 0.005$ | $0.925 \pm 0.002$ |
| SEFA (ours) | $\mathbf{0.904 \pm 0.001}$ | $\mathbf{0.919 \pm 0.001}$ | $\mathbf{0.676 \pm 0.005}$ | $\mathbf{0.957 \pm 0.000}$ |

| Model | MNIST | Fashion MNIST | METABRIC | TCGA |
|---|---|---|---|---|
| ACFlow | $0.667 \pm 0.003$ | $0.652 \pm 0.002$ | $0.542 \pm 0.006$ | $0.711 \pm 0.009$ |
| DIME | $0.731 \pm 0.002$ | $0.703 \pm 0.002$ | $0.670 \pm 0.007$ | $0.805 \pm 0.002$ |
| EDDI | $0.572 \pm 0.003$ | $0.604 \pm 0.001$ | $0.563 \pm 0.011$ | $0.634 \pm 0.005$ |
| Fixed MLP | $0.708 \pm 0.001$ | $0.690 \pm 0.001$ | $0.685 \pm 0.003$ | $0.799 \pm 0.004$ |
| GDFS | $0.732 \pm 0.001$ | $0.692 \pm 0.002$ | $0.671 \pm 0.005$ | $0.797 \pm 0.002$ |
| GSMRL | $0.701 \pm 0.002$ | $0.683 \pm 0.001$ | $0.665 \pm 0.002$ | $0.781 \pm 0.003$ |
| Opportunistic RL | $0.740 \pm 0.000$ | $0.708 \pm 0.000$ | $0.706 \pm 0.004$ | $0.838 \pm 0.002$ |
| Random | $0.661 \pm 0.001$ | $0.648 \pm 0.001$ | $0.647 \pm 0.005$ | $0.753 \pm 0.003$ |
| VAE | $0.716 \pm 0.001$ | $0.685 \pm 0.001$ | $0.690 \pm 0.004$ | $0.800 \pm 0.003$ |
| SEFA (ours) | $\mathbf{0.761 \pm 0.001}$ | $\mathbf{0.721 \pm 0.000}$ | $\mathbf{0.709 \pm 0.003}$ | $\mathbf{0.843 \pm 0.002}$ |

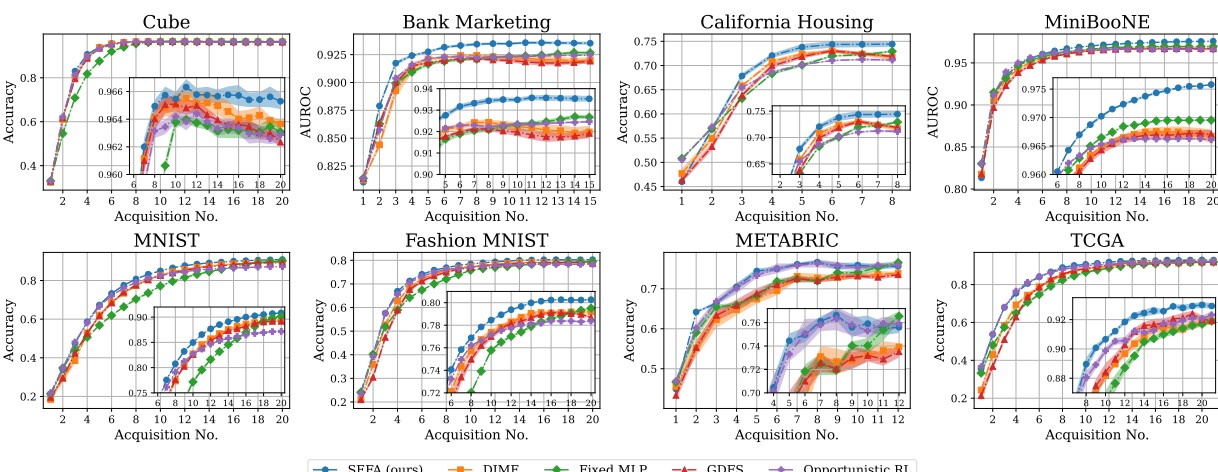

*Figure 3.* Evaluation metrics plots, starting from the first to the final acquisition across all datasets. Zoomed-in curves are shown in the bottom right corner of each plot.

tion metric through the acquisition curve, in particular on Bank Marketing and California Housing. Opportunistic RL, DIME, and GDFS perform approximately as well as each other across the real data. Interestingly, on MiniBooNE, the fixed ordering is the second best, despite other methods actively acquiring features. Again, the generative models underperform due to inaccurate CMI estimation.

**Image Classification.** Next we consider MNIST (LeCun et al., 1998) and Fashion MNIST (Xiao et al., 2017), and acquire up to twenty pixels. Here, the fixed ordering is inadequate, and the active methods perform better. Opportunistic RL outperforms DIME and GDFS, demonstrating RL is still an effective method for AFA despite its training difficulties, whereas the problems associated with CMI maximization are more fundamental. Again, SEFA outperforms

all methods by a significant margin, both in terms of average acquisition performance and the acquisition curve being consistently the highest throughout the acquisition.

### 6.3. Cancer Classification

Finally, we look at SEFA in the context of medicine. We consider two cancer classification tasks. The first is METABRIC (Curtis et al., 2012; Pereira et al., 2016), where the task is to predict the PAM50 status of breast cancer subjects from gene expression data. The six classes are Luminal A, Luminal B, HER2 Enriched, Basal Like, Claudin Low, and Normal Like. The second dataset uses The Cancer Genome Atlas (TCGA) (Weinstein et al., 2013). The goal is to predict the location of a tumor based on DNA methylation data. The average accuracies are given in Table 2 and the

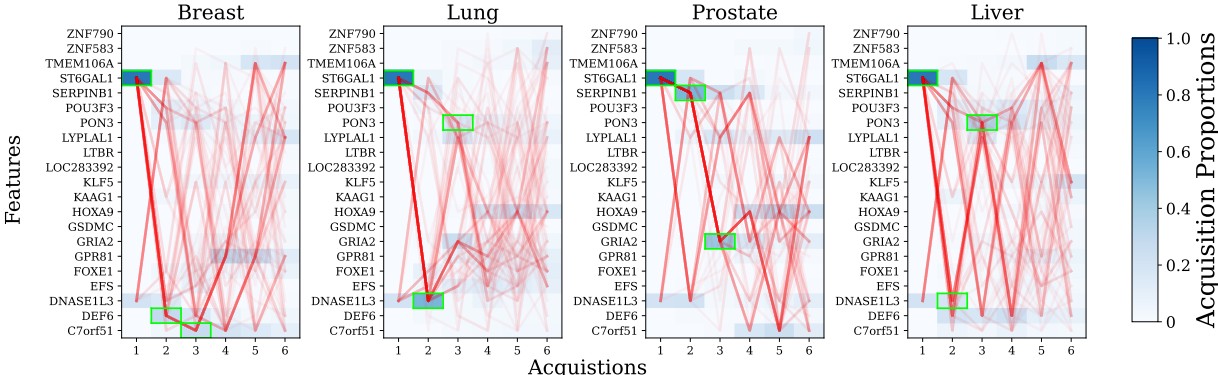

*Figure 4.* TCGA acquisition heat maps and trajectories for four tumor locations. We show the first six acquisitions. Notable acquisitions are highlighted with green boxes and discussed in Section 6.3. Despite the associated noise of a medical dataset, the trajectories and heat maps show instance-wise acquisitions.

acquisition curves in Figure 3. On METABRIC, SEFA and Opportunistic RL perform similarly, outperforming all other baselines. On TCGA, SEFA significantly outperforms all baselines, with Opportunistic RL a strong second, significantly outperforming DIME and GDFS, further demonstrating CMI is a flawed AFA objective.

To further validate the acquisitions of SEFA, we visualize the trajectories and heat maps for four cancer types in Figure 4, and provide scientific literature supporting the acquisitions made. Firstly, we can see that the acquisitions made are instance-wise, since the underlying acquisition heat maps change between classes. For example, HOXA9 (feature 9) is selected at acquisition 4, 5, and 6 for lung and prostate cancer, but not breast or liver. The first feature selected is almost always ST6GAL1 (feature 18), which is known to be upregulated in a number of cancers including breast, prostate, pancreatic, and ovarian (Garnham et al., 2019). For breast cancer, DEF6 (feature 2) is often acquired second, which has been identified to be correlated with metastatic behavior of breast cancer (Zhang et al., 2020). For lung and liver cancers, DNASE1L3 (feature 3) is often acquired second; this gene has been identified as a potential biomarker in liver and lung cancer (as well as breast, kidney, and stomach) (Deng et al., 2021). For prostate cancer, the second feature acquired tends to be SERPINB1 (feature 17), which is linked to prostate cancer (Lerman et al., 2019). For breast cancer, the third acquisition tends to be C7orf51 (feature 1), which is altered in triple negative invasive breast cancer (Brown, 2016). For lung and liver cancers, SEFA typically acquires PON3 (feature 15) as the third acquisition. It has been shown that PON3 is largely restricted to solid tumors such as those in liver, lung, and colon cancer (Schweikert et al., 2012). For prostate cancer, the most common third feature acquired is GRIA2 (feature 7), which has been found to correlate with the recurrence and prognosis of prostate cancer (Alwadi et al., 2022).

Note this is not an exhaustive list, and often it is helpful to acquire a feature that may not be associated with one type of cancer, if it is with another, since that feature helps to differentiate between them. Therefore, some acquisitions with a high proportion may not have an associated citation.

## 7. Conclusion

This paper considered Active Feature Acquisition, the test time task of actively choosing which features to observe to improve a prediction. We introduced SEFA, a novel approach for AFA by calculating the acquisition objective using samples from a suitably regularized stochastic latent space, moving away from previous solutions based on RL and CMI maximization. SEFA regularly outperformed previous methods across a range of tasks, and we validated acquired features in the scientific literature. Our ablation study demonstrated that each component of SEFA leads to performance gains.

**Limitations.** Currently SEFA applies to classification tasks but not to regression tasks. This is because SEFA requires separation of class probabilities during acquisition and this notion is not well defined for continuous labels. We view this as an interesting avenue for future work. One possible solution is to have two prediction heads, one that predicts the continuous label as a regression task, and one that predicts a discretized version of the label as a classification task. The classification head can be used for acquisition, and the regression head for prediction.

Additionally, due to requiring multiple latent samples, SEFA has larger memory requirements at inference time than RL baselines, depending on how many samples are used. However, CMI maximization methods with generative models also require multiple samples at inference time, so this is not a new limitation for AFA models.

## Acknowledgements

The results in the TCGA experiment are based upon data generated by the TCGA Research Network: `https://www.cancer.gov/tcga`. We thank the ICML reviewers for their comments and suggestions. We also thank Lucie Charlotte Magister and Iulia Duta for providing feedback. Alexander Norcliffe is supported by a GSK plc grant. Changhee Lee is supported by the National Research Foundation of Korea (NRF) grant funded by the Korea government (MSIT) (No. RS-2024-00358602) and the Institute of Information & Communications Technology Planning & Evaluation (IITP) grant funded by the Korea government (MSIT), Artificial Intelligence Graduate School Program (No. RS-2019-II190079, Korea University) and the Artificial Intelligence Star Fellowship Support Program to nurture the best talents (No. RS-2025-02304828).

## Impact Statement

Our paper is concerned with Active Feature Acquisition, the test time task of acquiring features to iteratively improve a model's predictions on a given data instance. Applications range from medical diagnosis to polling a population. We believe, on the whole, these applications pose a positive benefit to society. Naturally, since this is a general task (any task where features are not all available immediately), malicious applications do exist. For example, iteratively harvesting personal data to send targeted misinformation. However, this work does not focus on those applications, and since this area of machine learning research is still in its relative infancy, we do not envisage this occurring for the foreseeable future. An important consideration is the possibility of this work being used in a high-stakes setting or a scenario with ethical considerations, but producing incorrect or suboptimal results. In the medical scenario, a doctor might miss an important test to diagnose a patient, or conduct a painful/dangerous but unnecessary test. Currently, this work is not in a position to be deployed, so this is not a concern yet. However, if it were to be deployed, this problem can be mitigated by being used as a tool by domain experts to *aid* them in their decision making, and not replacing them.

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

## A. Additional Synthetic Heat Maps and Trajectories

To complement the synthetic experiments presented in Section 6, we provide the heat maps and trajectories for Syn 1 in Figure 5 and Syn 2 in Figure 6. In agreement with Table 1, SEFA clearly performs best on both Syn 1 and Syn 2. In both cases, $x_{11}$ is acquired first, informing the model where it needs to look next. All features are acquired by the theoretical minimum with the exception of a minority of trajectories. Opportunistic RL and DIME have a small but noticeable portion of sub-optimal trajectories on Syn 1 when $x_{11} < 0$. GDFS performs particularly poorly on Syn 1, when $x_{11} < 0$ a high proportion of required feature acquisitions are made after the theoretical minimum of 3 since initially $x_4$ and $x_5$ are selected. Additionally, GDFS regularly selects $x_{11}$ late into the acquisition process. On Syn 2, the three baselines do not place all attention on $x_{11}$ initially. In fact, Opportunistic RL and GDFS mostly acquire $x_7$ first since it provides the best immediate predictive signal. When $x_{11} \geq 0$, the baselines tend to acquire all relevant features in the theoretical minimum, albeit in sub-optimal orders. However, we see that when $x_{11} < 0$, this is not the case with many required acquisitions being made after the minimum of 3, since $x_7$ has been selected first.

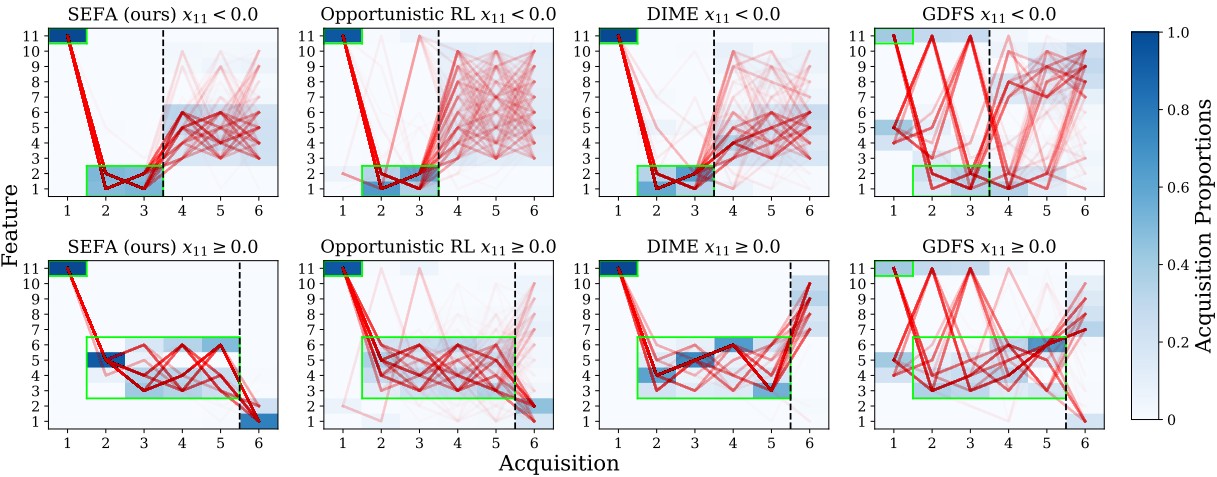

Figure 5. Acquisition heat maps and trajectories on Syn 1. Individual trajectories are plotted in red, with the acquisition proportions at each step as a heat map behind. We use green boxes to highlight the optimal strategy and a vertical black line to show the minimum number of features required (3 or 5).

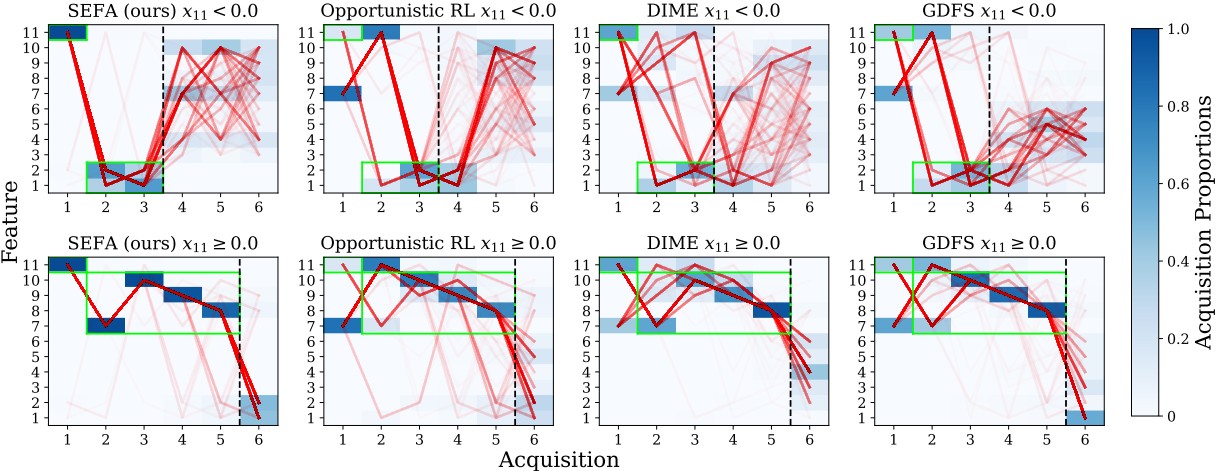

Figure 6. Acquisition heat maps and trajectories on Syn 2. Individual trajectories are plotted in red, with the acquisition proportions at each step as a heat map behind. We use green boxes to highlight the optimal strategy and a vertical black line to show the minimum number of features required (3 or 5).

# B. Synthetic Ablations and Sensitivity Analysis

**Heat maps and Trajectories.** We supplement the synthetic ablations in Table 1 by studying the acquisition heat maps and trajectories when we: calculate the acquisition objective in the feature space; use a deterministic encoder and use one acquisition sample of the latent space. We plot these for Syn 1-3 in Figures 7, 8 and 9. All three figures show that removing each of our proposed components degrades acquisition performance, confirming Table 1. All three reduced versions of SEFA, in all cases, select relevant features after the theoretical minimum since they regularly select $x_{11}$ late into the acquisition. Using the feature space almost never selects $x_{11}$ first. Acquiring with one latent sample leads to trajectories that approximately sample uniformly among all features relevant to a given synthetic task. Confirming that we need to take many acquisition samples to see a feature's effect on a diverse range of possible latent realizations.

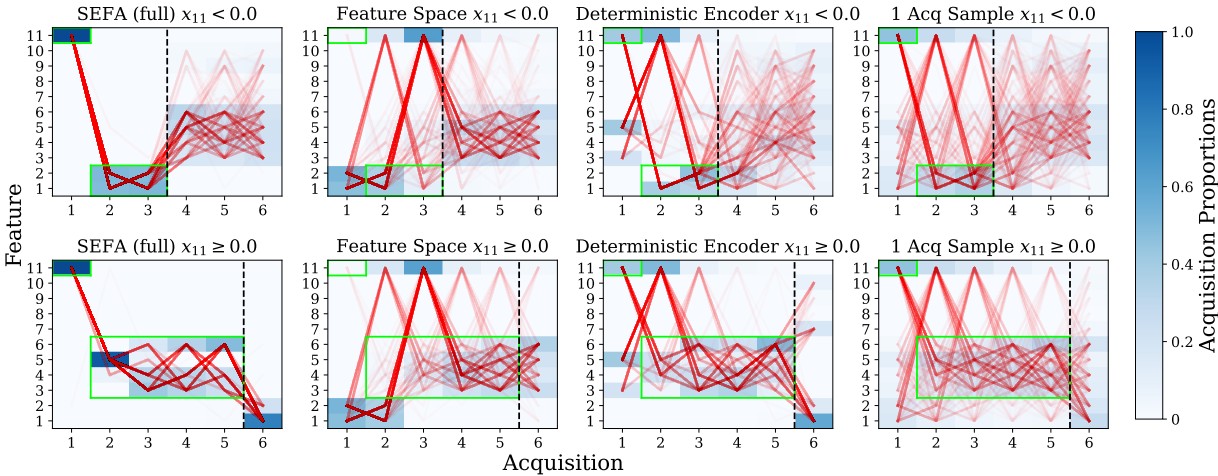

*Figure 7.* Acquisition heat maps and trajectories on Syn 1 ablations. Individual trajectories are plotted in red, with the acquisition proportions at each step as a heat map behind. We use green boxes to highlight the optimal strategy and a vertical black line to show the minimum number of features required (3 or 5).

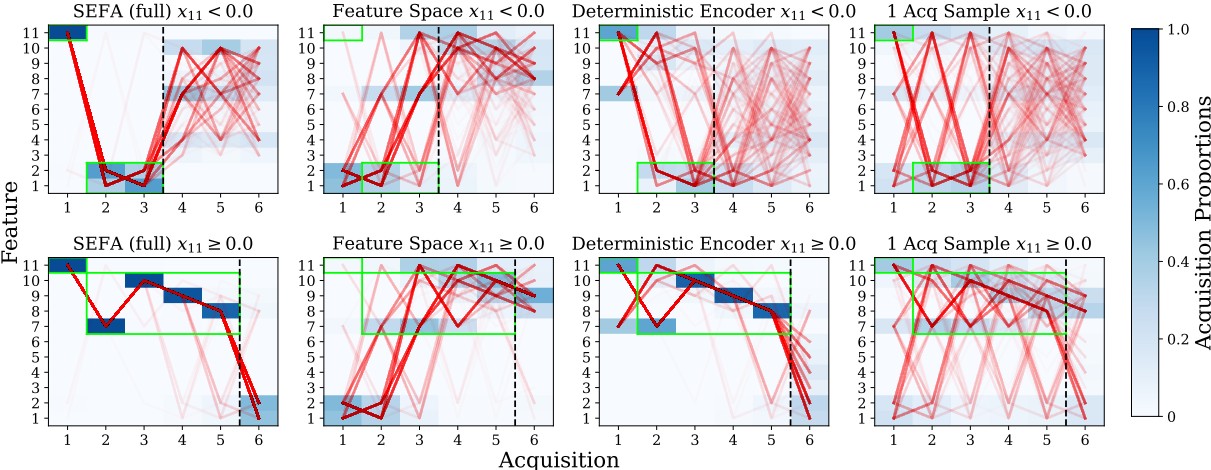

*Figure 8.* Acquisition heat maps and trajectories on Syn 2 ablations. Individual trajectories are plotted in red, with the acquisition proportions at each step as a heat map behind. We use green boxes to highlight the optimal strategy and a vertical black line to show the minimum number of features required (3 or 5).

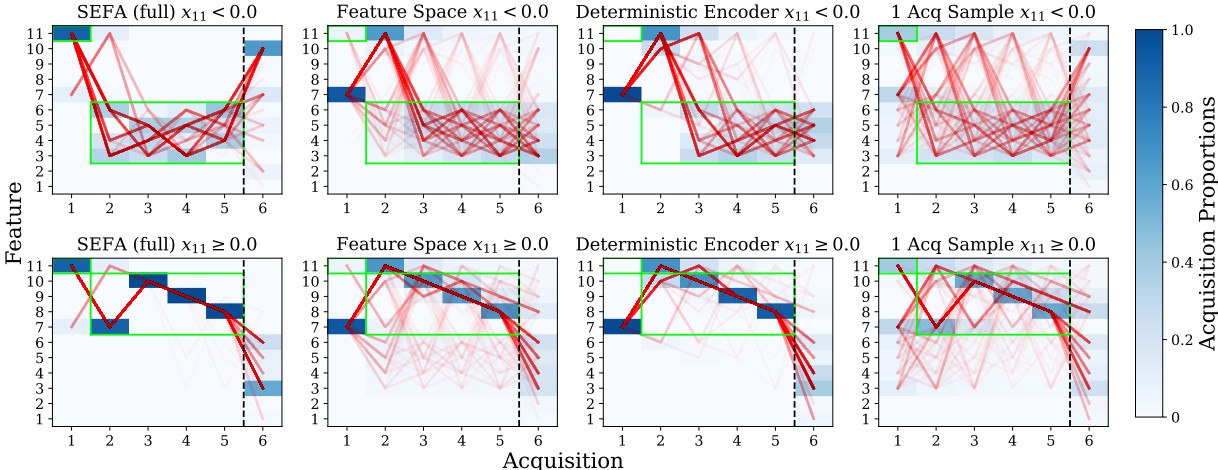

*Figure 9.* Acquisition heat maps and trajectories on Syn 3 ablations. Individual trajectories are plotted in red, with the acquisition proportions at each step as a heat map behind. We use green boxes to highlight the optimal strategy and a vertical black line to show the minimum number of features required (5).

**Sensitivity Analysis of $\beta$.** To further explore the importance of a well-regularized latent space, we conduct a sensitivity analysis on the hyperparameter $\beta$, keeping all other hyperparameters the same. Higher $\beta$ leads to the encoders removing more information about the features. We plot the number of acquisitions required to select all relevant features on the synthetic datasets in Figure 10. For all datasets, as expected, if $\beta$ is too high, the latent space is too heavily regularized. There is not enough label information in the latent space, so decisions made there lead to sub-optimal acquisitions. Equally, by not regularizing the latent space enough, there is nothing explicitly enforcing the latent space to remove irrelevant information about the features, also leading to sub-optimal acquisitions.

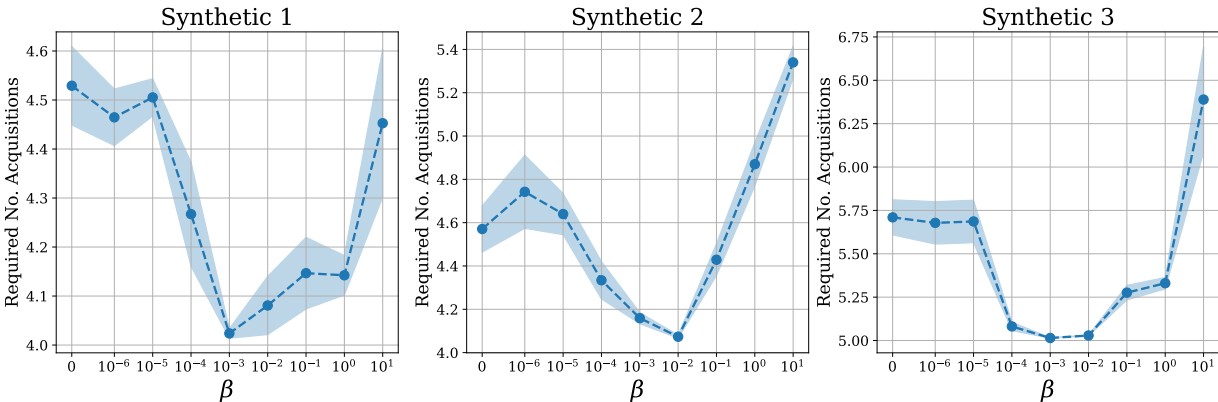

*Figure 10.* The number of acquisitions to select the relevant features for different values of $\beta$ on the synthetic tasks. The x-axis is logarithmic and includes zero.

**Sensitivity Analysis of Number of Acquisition Samples.** To further investigate the importance of using multiple acquisition samples to sample the full latent diversity, we run a sensitivity analysis on the synthetic tasks. We plot the number of acquisitions required to select all relevant features in Figure 11. As expected, if not enough samples are used, the number of acquisitions required is larger. We use 200 acquisition samples in our experiments, which is low enough for fast acquisition and high enough that performance has plateaued.

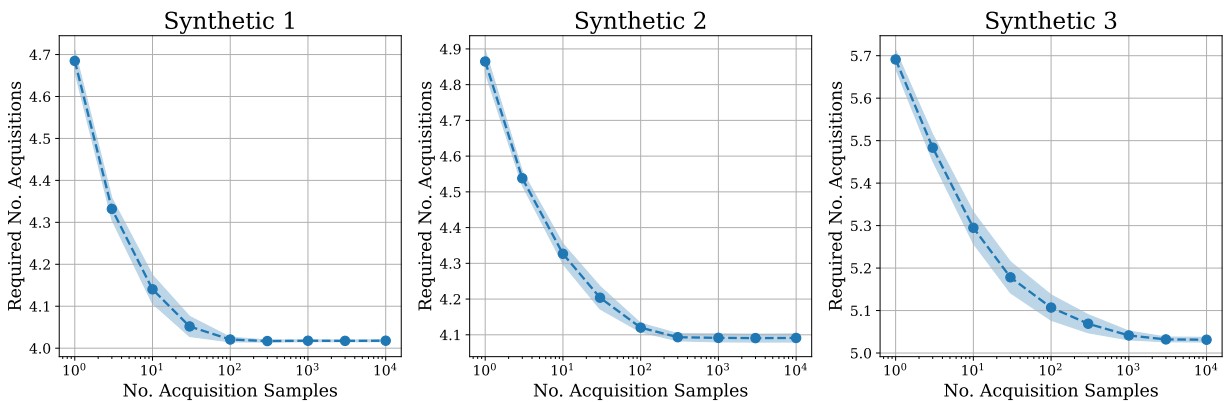

*Figure 11.* The number of acquisitions to select the relevant features for different numbers of acquisition samples on the synthetic tasks. The x-axis is logarithmic.

**Sensitivity Analysis of Number of Train Samples.** To further investigate the importance of using multiple training samples to encourage a diverse latent space, we run a sensitivity analysis on the synthetic tasks. We plot the number of acquisitions required to select all relevant features in Figure 12. For Syn 1 and Syn 2, we see that performance tends to improve with the number of samples as expected. For Syn 3, we see the best performance is achieved with 100 samples, which is the number we used in experiments.

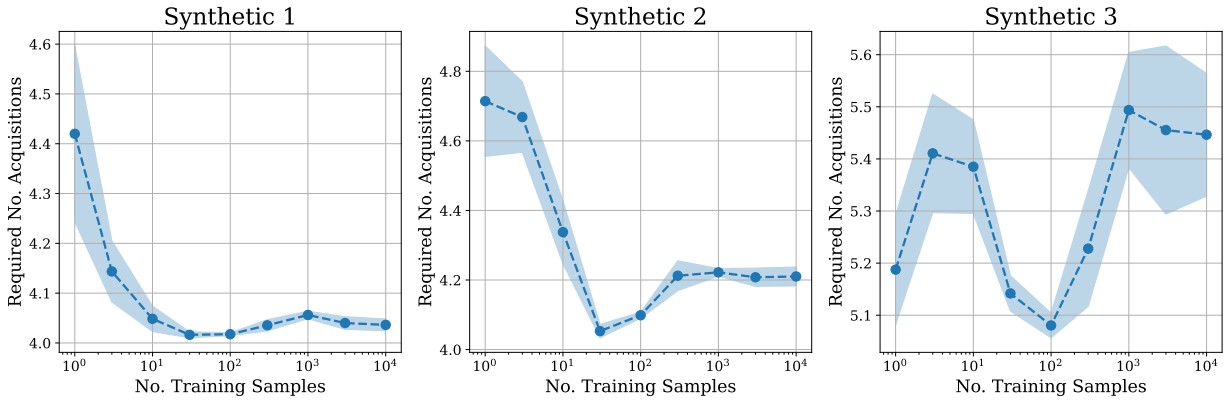

*Figure 12.* The number of acquisitions to select the relevant features for different numbers of training samples on the synthetic tasks. The x-axis is logarithmic.

**Sensitivity Analysis of Number of Latent Components.** To investigate the sensitivity of SEFA's performance to the number of latent components per feature, we run a sensitivity analysis on the synthetic tasks. We plot the number of acquisitions required to select all relevant features in Figure 13. We see the performance remains relatively constant, showing SEFA is fairly robust to this hyperparameter. We found between 5-10 is a good value such that there is enough capacity for rich representations of each feature, but not enough to overfit.

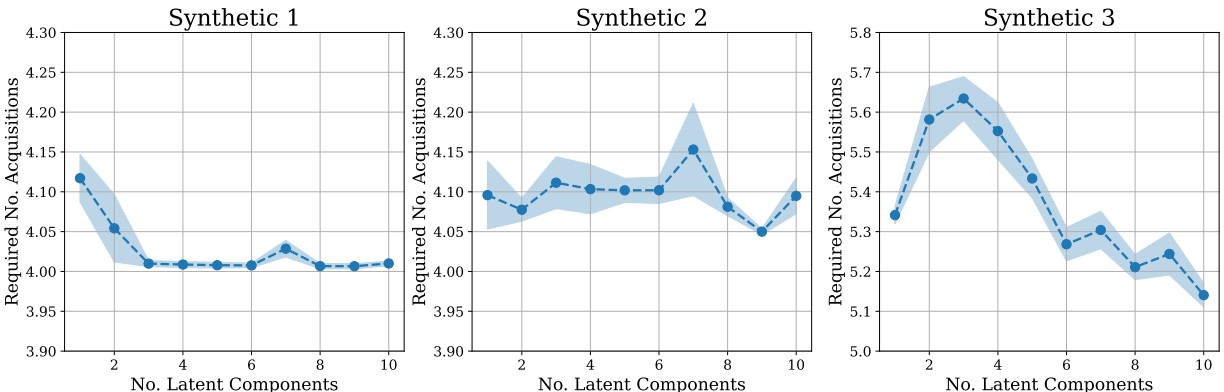

*Figure 13.* The number of acquisitions to select the relevant features for different numbers of latent components per feature on the synthetic tasks.

## C. Real Data Ablations

To further demonstrate that each novel model component leads to performance gains, we also carry out ablations on the real datasets. Additionally, here we investigate the final novelty we introduced, probability weighting, where we weight the scores during acquisition by the predicted probabilities $p_{\theta,\phi}(Y = c|\mathbf{x}_O)$. We investigate the use of this technique by removing the weight and taking a mean, treating each class equally. This was not meaningful on the synthetic ablations because this does not affect binary classification tasks. To see this, recall how features are scored in (2):

$$R(\mathbf{x}_O, i) = \sum_{c \in [C]} p_{\theta,\phi}(Y = c|\mathbf{x}_O) \, \mathbb{E}_{p_\theta(\mathbf{z}|\mathbf{x}_O)} \, r(c, \mathbf{z}, i).$$

Writing this in the binary case (class labels are either 0 or 1) gives

$$R(\mathbf{x}_O, i) = p_{\theta,\phi}(Y = 0|\mathbf{x}_O) \, \mathbb{E}_{p_\theta(\mathbf{z}|\mathbf{x}_O)} \, r(0, \mathbf{z}, i) + p_{\theta,\phi}(Y = 1|\mathbf{x}_O) \, \mathbb{E}_{p_\theta(\mathbf{z}|\mathbf{x}_O)} \, r(1, \mathbf{z}, i).$$

$p_\phi(Y = 1|\mathbf{z}) = 1 - p_\phi(Y = 0|\mathbf{z})$, $\nabla_{\mathbf{z}} p_\phi(Y = 1|\mathbf{z}) = -\nabla_{\mathbf{z}} p_\phi(Y = 0|\mathbf{z})$, therefore $r(0, \mathbf{z}, i) = r(1, \mathbf{z}, i)$, since the gradients point in opposite directions, and taking Euclidean norms and normalizing is agnostic to the negative sign. Therefore

$$R(\mathbf{x}_O, i) = p_{\theta,\phi}(Y = 0|\mathbf{x}_O) \, \mathbb{E}_{p_\theta(\mathbf{z}|\mathbf{x}_O)} \, r(0, \mathbf{z}, i) + p_{\theta,\phi}(Y = 1|\mathbf{x}_O) \, \mathbb{E}_{p_\theta(\mathbf{z}|\mathbf{x}_O)} \, r(0, \mathbf{z}, i),$$

$$R(\mathbf{x}_O, i) = \big(p_{\theta,\phi}(Y = 0|\mathbf{x}_O) + p_{\theta,\phi}(Y = 1|\mathbf{x}_O)\big) \, \mathbb{E}_{p_\theta(\mathbf{z}|\mathbf{x}_O)} \, r(0, \mathbf{z}, i),$$

$$R(\mathbf{x}_O, i) = \mathbb{E}_{p(\mathbf{z}|\mathbf{x}_O)} \, r(0, \mathbf{z}, i) = \mathbb{E}_{p_\theta(\mathbf{z}|\mathbf{x}_O)} \, r(1, \mathbf{z}, i).$$

The weighting is removed in the binary case, proving probability weighting only affects the multi-class setting. We run the ablations on the real datasets as well as Cube, where the precise ordering is not known a priori. We provide average evaluation metrics during acquisition in Table 3. Calculating the acquisition objective in feature space consistently leads to the worst result (except for Cube and METABRIC where it is second worst), demonstrating that calculating the objective in latent space is a crucial component to the performance of SEFA, even if we give up the explicit ability to model conditional distributions between the features. Following this, using a deterministic encoder or one acquisition sample is the next worst result, showing the second novelty of using stochastic encoders to consider a diverse set of possible unobserved latent realizations is a key factor contributing to the performance of SEFA. Removing probability weighting, as expected, has no effect on binary classification tasks (Bank Marketing and MiniBooNE), it does however have a significant effect on MNIST, Fashion MNIST, METABRIC and TCGA where there are multiple classes. The performance is also marginally worse without probability weighting on California Housing, and does not change on Cube. In two cases, removing regularization improves performance, but in six out of eight cases and for the synthetic datasets, non-zero $\beta$ improves the acquisition performance of SEFA. Using one train sample consistently leads to slightly worse performance, matching the results on the synthetic datasets. Removing the normalization in the $r$ calculation has a minimal effect: it improves performance on Cube,

MNIST, and TCGA (the change is marginal in these cases) and worsens performance on Fashion MNIST, showing the exact form for $r$ is not as important in the acquisition objective as using stochastic latent encodings or probability weighting.

An additional ablation that does not apply to the synthetic datasets is the use of a copula transform (WO Copula). It transforms the continuous features using the empirical cumulative distribution function, rather than raw feature values. It does not affect the synthetic datasets because those features are sampled from a standard normal. Note the transform is *specific* to SEFA due to its information bottleneck regularization (see Appendix I.2). Similar to removing normalization, on some datasets the performance improves slightly by removing the Copula (Cube, MNIST, METABRIC), and for others the performance is worse (Bank Marketing, MiniBooNE, Fashion MNIST). The difference is marginal compared to using the feature space, deterministic encoders, or removing probability weighting, the main novelties of our work.

We plot acquisition curves for a key subset of ablations in Figure 14. We do not show the feature space calculation trajectory since it is consistently the worst (see Table 3). Deterministic encoders (green) regularly perform noticeably worse than using stochastic encoders. Removing probability weighting (orange) also leads to significantly worse performance on four of the six multi-class datasets. Using multiple latent acquisition and training samples is also required for the best performance.

*Table 3.* Average acquisition metrics on ablations. We give the mean and standard error. Ablations that outperform SEFA are underlined.

| Ablation | Cube | Bank Marketing | California Housing | MiniBooNE |
|---|---|---|---|---|
| $\beta = 0$ | $0.906 \pm 0.000$ | $0.917 \pm 0.001$ | $0.680 \pm 0.005$ | $0.957 \pm 0.000$ |
| 1 Acquisition Sample | $0.889 \pm 0.001$ | $0.912 \pm 0.001$ | $0.667 \pm 0.004$ | $0.952 \pm 0.000$ |
| 1 Train Sample | $0.901 \pm 0.002$ | $0.912 \pm 0.003$ | $0.676 \pm 0.006$ | $0.953 \pm 0.000$ |
| Deterministic Encoder | $0.900 \pm 0.000$ | $0.893 \pm 0.006$ | $0.667 \pm 0.005$ | $0.944 \pm 0.002$ |
| Feature Space Calculation | $0.897 \pm 0.002$ | $0.880 \pm 0.003$ | $0.600 \pm 0.004$ | $0.914 \pm 0.004$ |
| WO Copula | $0.905 \pm 0.000$ | $0.917 \pm 0.002$ | $0.675 \pm 0.002$ | $0.952 \pm 0.001$ |
| WO Normalization | $0.905 \pm 0.000$ | $0.919 \pm 0.001$ | $0.676 \pm 0.005$ | $0.957 \pm 0.000$ |
| WO Prob Weighting | $0.904 \pm 0.001$ | $0.919 \pm 0.001$ | $0.674 \pm 0.005$ | $0.957 \pm 0.000$ |
| SEFA (full) | $0.904 \pm 0.001$ | $0.919 \pm 0.001$ | $0.676 \pm 0.005$ | $0.957 \pm 0.000$ |
| Ablation | MNIST | Fashion MNIST | METABRIC | TCGA |
| $\beta = 0$ | $0.759 \pm 0.000$ | $0.719 \pm 0.000$ | $0.706 \pm 0.003$ | $0.842 \pm 0.002$ |
| 1 Acquisition Sample | $0.728 \pm 0.000$ | $0.699 \pm 0.000$ | $0.694 \pm 0.003$ | $0.827 \pm 0.002$ |
| 1 Train Sample | $0.741 \pm 0.002$ | $0.707 \pm 0.001$ | $0.705 \pm 0.003$ | $0.834 \pm 0.002$ |
| Deterministic Encoder | $0.741 \pm 0.002$ | $0.704 \pm 0.001$ | $0.702 \pm 0.004$ | $0.838 \pm 0.002$ |
| Feature Space Calculation | $0.697 \pm 0.001$ | $0.634 \pm 0.001$ | $0.695 \pm 0.001$ | $0.795 \pm 0.001$ |
| WO Copula | $0.762 \pm 0.001$ | $0.717 \pm 0.001$ | $0.710 \pm 0.002$ | $0.843 \pm 0.002$ |
| WO Normalization | $0.762 \pm 0.001$ | $0.718 \pm 0.001$ | $0.709 \pm 0.002$ | $0.844 \pm 0.002$ |
| WO Prob Weighting | $0.751 \pm 0.001$ | $0.694 \pm 0.001$ | $0.698 \pm 0.003$ | $0.832 \pm 0.001$ |
| SEFA (full) | $0.761 \pm 0.001$ | $0.721 \pm 0.000$ | $0.709 \pm 0.003$ | $0.843 \pm 0.002$ |

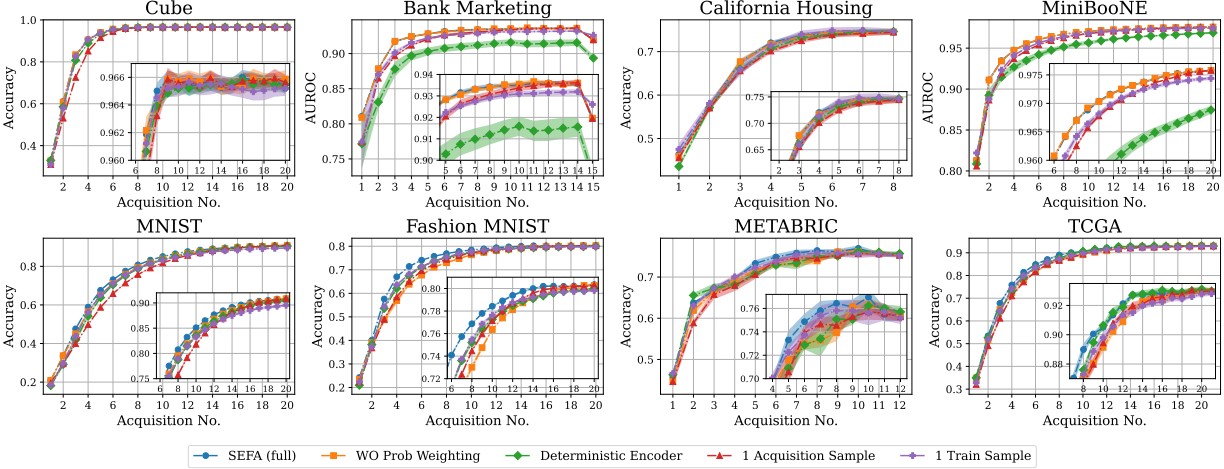

*Figure 14.* Evaluation metrics starting from the first to the final acquisition for the ablations. To distinguish curves, we provide zoomed-in versions of the plots in the bottom right corner of each one.

## D. Additional Experiment: Robustness to Noise

To investigate how robust SEFA is to noisy features, we repeat the Syn 1-3 experiments; however, after calculating the label, we add normally distributed noise to the features, with increasing standard deviations. The results are given in Table 4. All models get worse with more noise. SEFA remains the best model at all levels of noise. This is likely due to the latent space regularization removing noise between the feature space and the latent space.

*Table 4.* Number of acquisitions to acquire the correct features under increasing levels of noise, the lower the better. We provide the mean and one standard error.

| Syn 1 | $\sigma = 0.0$ | $\sigma = 0.1$ | $\sigma = 0.2$ | $\sigma = 0.4$ |
|---|---|---|---|---|
| ACFlow | $7.730 \pm 0.139$ | $7.754 \pm 0.199$ | $7.755 \pm 0.275$ | $8.255 \pm 0.131$ |
| DIME | $4.079 \pm 0.064$ | $4.688 \pm 0.211$ | $4.703 \pm 0.276$ | $5.368 \pm 0.267$ |
| EDDI | $9.197 \pm 0.203$ | $8.988 \pm 0.285$ | $9.216 \pm 0.136$ | $9.382 \pm 0.245$ |
| Fixed MLP | $6.009 \pm 0.000$ | $6.312 \pm 0.202$ | $7.321 \pm 0.378$ | $6.009 \pm 0.000$ |
| GDFS | $4.568 \pm 0.219$ | $4.566 \pm 0.187$ | $4.543 \pm 0.164$ | $5.583 \pm 0.283$ |
| GSMRL | $5.570 \pm 0.127$ | $5.495 \pm 0.126$ | $5.778 \pm 0.120$ | $6.858 \pm 0.269$ |
| Opportunistic RL | $4.201 \pm 0.041$ | $4.347 \pm 0.091$ | $4.720 \pm 0.142$ | $5.488 \pm 0.084$ |
| Random | $9.484 \pm 0.006$ | $9.495 \pm 0.010$ | $9.495 \pm 0.010$ | $9.495 \pm 0.010$ |
| VAE | $6.589 \pm 0.088$ | $6.866 \pm 0.041$ | $7.005 \pm 0.086$ | $7.095 \pm 0.078$ |
| SEFA (ours) | $\mathbf{4.017 \pm 0.003}$ | $\mathbf{4.100 \pm 0.004}$ | $\mathbf{4.207 \pm 0.008}$ | $\mathbf{4.406 \pm 0.004}$ |
| Syn 2 | $\sigma = 0.0$ | $\sigma = 0.1$ | $\sigma = 0.2$ | $\sigma = 0.4$ |
| ACFlow | $7.527 \pm 0.254$ | $7.366 \pm 0.325$ | $7.927 \pm 0.363$ | $7.974 \pm 0.072$ |
| DIME | $4.581 \pm 0.217$ | $4.830 \pm 0.057$ | $5.093 \pm 0.218$ | $5.015 \pm 0.124$ |
| EDDI | $9.214 \pm 0.415$ | $9.550 \pm 0.342$ | $9.079 \pm 0.244$ | $9.571 \pm 0.225$ |
| Fixed MLP | $5.996 \pm 0.000$ | $6.693 \pm 0.373$ | $6.095 \pm 0.100$ | $6.494 \pm 0.315$ |
| GDFS | $4.484 \pm 0.159$ | $4.981 \pm 0.211$ | $5.655 \pm 0.386$ | $6.804 \pm 0.384$ |
| GSMRL | $6.227 \pm 0.185$ | $6.283 \pm 0.146$ | $6.525 \pm 0.158$ | $7.286 \pm 0.088$ |
| Opportunistic RL | $4.846 \pm 0.021$ | $5.042 \pm 0.007$ | $5.108 \pm 0.064$ | $5.246 \pm 0.041$ |
| Random | $9.499 \pm 0.005$ | $9.504 \pm 0.007$ | $9.504 \pm 0.007$ | $9.504 \pm 0.007$ |
| VAE | $6.667 \pm 0.140$ | $6.874 \pm 0.161$ | $7.057 \pm 0.134$ | $7.204 \pm 0.094$ |
| SEFA (ours) | $\mathbf{4.099 \pm 0.009}$ | $\mathbf{4.247 \pm 0.042}$ | $\mathbf{4.388 \pm 0.045}$ | $\mathbf{4.778 \pm 0.170}$ |
| Syn 3 | $\sigma = 0.0$ | $\sigma = 0.1$ | $\sigma = 0.2$ | $\sigma = 0.4$ |
| ACFlow | $9.194 \pm 0.278$ | $9.105 \pm 0.218$ | $9.184 \pm 0.248$ | $9.284 \pm 0.231$ |
| DIME | $5.667 \pm 0.038$ | $5.771 \pm 0.222$ | $6.180 \pm 0.161$ | $6.213 \pm 0.180$ |
| EDDI | $9.794 \pm 0.186$ | $9.969 \pm 0.156$ | $9.906 \pm 0.212$ | $9.710 \pm 0.136$ |
| Fixed MLP | $7.999 \pm 0.000$ | $7.999 \pm 0.000$ | $7.999 \pm 0.000$ | $7.999 \pm 0.000$ |
| GDFS | $5.587 \pm 0.201$ | $6.839 \pm 0.338$ | $7.179 \pm 0.134$ | $7.814 \pm 0.349$ |
| GSMRL | $8.199 \pm 0.067$ | $8.326 \pm 0.233$ | $8.347 \pm 0.122$ | $8.627 \pm 0.186$ |
| Opportunistic RL | $5.850 \pm 0.072$ | $5.859 \pm 0.063$ | $6.135 \pm 0.114$ | $6.672 \pm 0.105$ |
| Random | $9.987 \pm 0.008$ | $9.991 \pm 0.006$ | $9.991 \pm 0.006$ | $9.991 \pm 0.006$ |
| VAE | $7.888 \pm 0.065$ | $7.968 \pm 0.068$ | $8.042 \pm 0.102$ | $8.469 \pm 0.064$ |
| SEFA (ours) | $\mathbf{5.084 \pm 0.026}$ | $\mathbf{5.448 \pm 0.042}$ | $\mathbf{5.689 \pm 0.061}$ | $\mathbf{6.207 \pm 0.058}$ |

## E. Additional TCGA Trajectories

To further augment the TCGA analysis in Section 6, we provide the heat maps and trajectories across all 17 tumor locations in Figure 15. We see the selections are instance-wise orderings since different classes have different heat maps and trajectories (as well as different trajectories emerging for the same tumor location). Due to the nature of the task and data, there is still associated noise. Further to the justification in the main paper, we see that in many cases DNASE1L3 (feature 3) is selected after ST6GAL1 (feature 18). This is because it has been linked to: bladder cancer, breast cancer, gastric carcinoma, liver cancer, lung adenocarcinoma, lung squamous cell carcinoma, ovarian cancer, cervical squamous cell carcinoma, head-neck squamous cell carcinoma, pancreatic adenocarcinoma, and kidney renal clear cell carcinoma (Deng et al., 2021). Additionally, it has been linked to colon cancer progression (Li et al., 2023), and was found to be downregulated in prostate adenocarcinoma and uterine corpus endometrial carcinoma (Deng et al., 2021). DNASE1L3 is regularly selected for all of these tumor locations, even first occasionally, since it is a strong predictor on its own. We likely see the selection appearing for brain and bone marrow as a way to rule out these other likely locations after selecting ST6GAL1. Additional notable acquisitions with supporting literature are: kidney cancer acquiring POU3F3 (feature 16) (Zhang et al., 2021), KAAG1 (feature 10) (the name is Kidney Associated Antigen 1), and HOXA9 (feature 9) (Shenoy et al., 2024); colon cancer acquiring HOXA9 (Cui et al., 2024) and thyroid cancer acquiring FOXE1 (feature 5) (Penna-Martinez et al., 2014). Note this is not an exhaustive list.

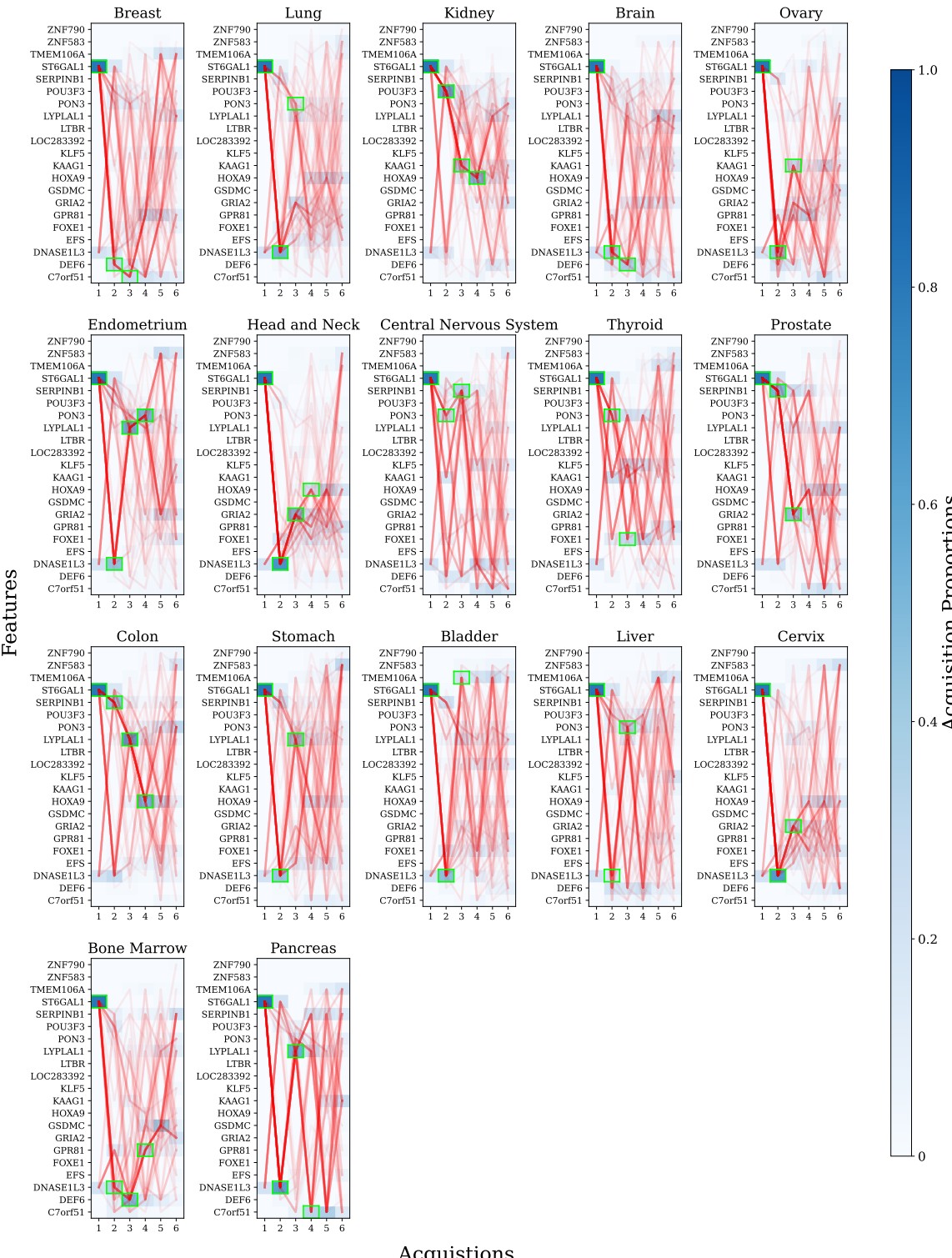

*Figure 15.* Acquisition trajectories for TCGA across all classes. Individual trajectories are given in red, with the heat map of acquisition proportions at each step shown behind. Notable acquisitions with high proportions are highlighted with green boxes.

# F. Indicator Example

Here we elaborate on our indicator example, a simple case where CMI fails. First, we demonstrate that CMI fails and then we show that, by considering possible unobserved feature values in the calculation, we can recover the optimal policy.

Recall the example, we have features $X \in \{0, 1\}^d \times [d]$, i.e the first $d$ dimensions are binary and the final feature is an indicator. The label is given by the value of the feature indexed by the indicator $y = x_{x_{d+1}}$. In the absence of any of the first $d$ features, the indicator and label are independent $p(y, x_{d+1}) = p(y)p(x_{d+1})$. Substituting this into the definition of mutual information gives

$$I(Y; X_{d+1}) = D_{\mathrm{KL}}(p(Y, X_{d+1})||p(Y)p(X_{d+1})) = D_{\mathrm{KL}}(p(Y)p(X_{d+1})||p(Y)p(X_{d+1})) = 0.$$

Now consider the mutual information for the other features. Due to the symmetry of the problem, the mutual information for one of these features is the same for all others. The mutual information can be more usefully written as

$$I(Y; X_i) = H(Y) - \int H(Y|x_i)p(x_i)dx_i.$$

The entropy of the label is $\log 2$ since there is equal chance of being 0 or 1. Again, using the symmetry of the system, the entropy of $Y$ if $X_i = 0$ is the same as if $X_i = 1$, so we only calculate for one case. When $X_i = 0$, the probability of $Y = 0$ is $\frac{1}{d} \times 1 + \frac{d-1}{d} \times \frac{1}{2}$, since in $\frac{1}{d}$ cases it takes the exact value of $X_i$ based on the value of the indicator, and in $\frac{d-1}{d}$ cases $Y$ is given by a different unknown feature value. This gives $p(Y = 0|X_i = 0) = \frac{d+1}{2d}$. The expression for binary entropy, $-p\log(p) - (1-p)\log(1-p)$ is maximized by $p = 0.5$, giving $\log 2$. Since $p(Y = 0|X_i = 0) > 0.5$, the entropy is lower than $\log 2$ in this case. Exploiting the symmetry of the system we conclude that $\int H(Y|x_i)p(x_i)dx_i < \log 2$, and therefore $I(Y; X_i) > 0$.

Therefore, the indicator is never chosen first, which is a sub-optimal strategy. It can be shown, but is not necessary, that the indicator will be chosen second. A sketch of the reasoning is that now that the value of one feature is known, the indicator and the label are correlated. Therefore, there is non-zero CMI, which turns out to be larger than for the other features, and once the indicator is chosen, the correct feature is the only feature afterward with non-zero CMI. So this strategy will acquire the correct features in 3 selections $\frac{d-1}{d}$ of the time (random feature, indicator, correct feature) and in 2 selections $\frac{1}{d}$ of the time (correct feature, indicator). Thus, the expected number of acquisitions for this strategy is

$$2\frac{1}{d} + 3\frac{d-1}{d} = 3 - \frac{1}{d}$$

So as $d$ gets large, the expected number of required acquisitions approaches 3.

Now consider the adjusted solution of using an information-theoretic objective that considers the values of other features. Recall Proposition 4.2, we propose $\mathbb{E}_{p(\mathbf{x}_U|\mathbf{x}_O)} I(X_i; Y|\mathbf{x}_O, \mathbf{x}_U)$ recovers the optimal strategy, where $\mathbf{x}_U$ is the vector of all other unobserved features. We prove that this will lead to an optimal strategy below.

Initially there are no features, so the acquisition objective is $\int I(X_i; Y|\mathbf{x}_U)p(\mathbf{x}_U)d\mathbf{x}_U$. Writing this in terms of entropies gives

$$\int I(X_i; Y|\mathbf{x}_U)p(\mathbf{x}_U)d\mathbf{x}_U = \int \left( H(Y|\mathbf{x}_U) - \int H(Y|x_i, \mathbf{x}_U)p(x_i|\mathbf{x}_U)dx_i \right) p(\mathbf{x}_U)d\mathbf{x}_U.$$

The entropy when all features are known is zero, so for any $i$ the objective simplifies to

$$\int H(Y|\mathbf{x}_U)p(\mathbf{x}_U)d\mathbf{x}_U.$$

If we consider the first $d$ features, we can again apply symmetry to calculate this quantity for one feature - feature 1 - and apply it to all of them. In $\frac{d-1}{d}$ cases the entropy is zero, since we will have all of the information required. However if $x_{d+1} = 1$, then $H(Y|\mathbf{x}_U) = \log 2$, since we don't know the value of feature 1 and therefore $Y$ has equal likelihood of being 0 or 1, this happens in $\frac{1}{d}$ cases so this quantity is $\frac{\log 2}{d}$ for the first $d$ features.

For the indicator, $p(Y = 0|\mathbf{x}_U)$ is the proportion of the first $d$ features that are 0 for a given sample $\mathbf{x}_U$. All features are independent with probability 0.5 of being 0, so the expression is given by a binomial distribution with $d$ trials

$$\sum_{k=0}^{d} \binom{d}{k} \frac{1}{2^d} \left( -\left(\frac{k}{d}\right) \log\left(\frac{k}{d}\right) - \left(1 - \frac{k}{d}\right) \log\left(1 - \frac{k}{d}\right) \right).$$

It is not immediately clear that this is larger than the quantity $\frac{\log 2}{d}$ for the other features. The first thing we can do is calculate this quantity when $d = 3$, which gives $0.477$, and this is larger than $\frac{\log 2}{3} = 0.231$. And the next thing is to notice that this quantity is increasing with $d$, since as $d$ gets larger there will be more probability mass at $k = \frac{d}{2}$. As $d \to \infty$ the binomial distribution becomes Gaussian with mean $\frac{d}{2}$ and variance $\frac{d}{4}$, so $\frac{k}{d}$ will approximately be distributed normally with mean $\frac{1}{2}$ and standard deviation $\frac{1}{2\sqrt{d}}$. Therefore this quantity asymptotes towards $\log 2$.

Therefore for $d \geq 3$, this objective will choose the indicator first, and not the other features (for $d = 2$ all features are scored the same, and for $d = 1$ the indicator is not the optimal choice). After choosing the indicator, the second selection is trivial. The relevant feature has non-zero CMI, all other features are independent of the label conditioned on the indicator so they have zero CMI. Therefore the correct feature is chosen. This strategy's expected number of acquisitions is 2, which is less than $3 - \frac{1}{d}$.

This example illustrates that by considering the possible realizations in the calculation, and not marginalizing them out, we can make long-term acquisitions. Note we do not use this specific quantity in our paper since it involves an additional expectation over unobserved values as well as the expectation inside the CMI, which is intractable. This does not even account for the difficulty in estimating the conditional distributions in feature space.

## G. Entropy Example

In Section 4, we claimed that CMI maximization can lead to acquisitions that focus on making low probabilities lower, rather than distinguishing between possible answers. Here we provide a concrete example of this occurring.

Consider a feature vector with two features, $X \in \{1, 2, 3\}^2$, and a label with three possible classes, $Y \in \{1, 2, 3\}$. To generate the label from the features, we start with a vector with three zeros $\mathbf{v} = [0.0, 0.0, 0.0]$. The value of $x_1$ tells us which value in $\mathbf{v}$ to reduce by $8$. And the value of $x_2$ tells us which value in $\mathbf{v}$ to increase by $6.5$. For example if $\mathbf{x} = [1, 2]$, then $\mathbf{v} = [-8.0, 6.5, 0.0]$. $p(y|\mathbf{x})$ is then given by the softmax of $\mathbf{v}$. With these rules, we can calculate all possible feature values and probabilities in Table 5.

*Table 5.* All feature values and corresponding $y$ probabilities given by the problem in Appendix G.

| $x_1$ | $x_2$ | $v_1$ | $v_2$ | $v_3$ | $p(Y=1|\mathbf{x})$ | $p(Y=2|\mathbf{x})$ | $p(Y=3|\mathbf{x})$ |
|---|---|---|---|---|---|---|---|
| 1 | 1 | $-1.5$ | 0.0 | 0.0 | 0.10037 | 0.44982 | 0.44982 |
| 1 | 2 | $-8.0$ | 6.5 | 0.0 | 0.00000 | 0.99850 | 0.00150 |
| 1 | 3 | $-8.0$ | 0.0 | 6.5 | 0.00000 | 0.00150 | 0.99850 |
| 2 | 1 | 6.5 | $-8.0$ | 0.0 | 0.99850 | 0.00000 | 0.00150 |
| 2 | 2 | 0.0 | $-1.5$ | 0.0 | 0.44982 | 0.10037 | 0.44982 |
| 2 | 3 | 0.0 | $-8.0$ | 6.5 | 0.00150 | 0.00000 | 0.99850 |
| 3 | 1 | 6.5 | 0.0 | $-8.0$ | 0.99850 | 0.00150 | 0.00000 |
| 3 | 2 | 0.0 | 6.5 | $-8.0$ | 0.00150 | 0.99850 | 0.00000 |
| 3 | 3 | 0.0 | 0.0 | $-1.5$ | 0.44982 | 0.44982 | 0.10037 |

By taking the average, or by seeing that all components of $\mathbf{v}$ are treated equally across all feature values, the marginal of the label is $p(y) = [0.33, 0.33, 0.33]$, and the entropy is $H(Y) = \log 3$. Based on this, we can then calculate the marginal distributions and the entropies of the marginals given in Table 6.

*Table 6.* Marginal distributions and entropies calculated using Table 5.

| $x_1$ | $x_2$ | $p(Y=1|\mathbf{x})$ | $p(Y=2|\mathbf{x})$ | $p(Y=3|\mathbf{x})$ | $H(Y|\mathbf{x})$ | $H(Y) - H(Y|\mathbf{x})$ |
|---|---|---|---|---|---|---|
| 1 | Missing | 0.03346 | 0.48327 | 0.48327 | 0.81652 | 0.28210 |
| 2 | Missing | 0.48327 | 0.03346 | 0.48327 | 0.81652 | 0.28210 |
| 3 | Missing | 0.48327 | 0.48327 | 0.03346 | 0.81652 | 0.28210 |
| Missing | 1 | 0.69912 | 0.15044 | 0.15044 | 0.82016 | 0.27845 |
| Missing | 2 | 0.15044 | 0.69912 | 0.15044 | 0.82016 | 0.27845 |
| Missing | 3 | 0.15044 | 0.15044 | 0.69912 | 0.82016 | 0.27845 |

Feature 2 is consistently more useful for identifying the most likely class, since it can always identify a class with likelihood $0.699$, and feature 1 is better at identifying which class is least likely. However, feature 1 consistently has a lower entropy, so it would be selected before feature 2 by CMI.

# H. Dataset Details

Here we provide all the details about each dataset, including sizes, number of features, and how to access the real datasets.

**Synthetic.** The synthetic experiments are based on (Yoon et al., 2019), where we know the features that are predictive, and we know that there is a heterogeneous order. The datasets are binary datasets where the feature vector has eleven independent features drawn from a standard normal. There are three possible logits:

$$\ell_1 = 4x_1x_2, \qquad \ell_2 = \sum_{i=3}^{6} 1.2x_i^2 - 4.2, \qquad \ell_3 = -10\sin(0.2x_7) + |x_8| + x_9 + e^{-x_{10}} - 2.4$$

Then, for a given logit value, the label is sampled from a Bernoulli distribution with probability $p(Y = 1) = 1/(1 + e^\ell)$. We construct three datasets:

- Synthetic 1: If $x_{11} < 0$ we use $\ell_1$, otherwise $\ell_2$
- Synthetic 2: If $x_{11} < 0$ we use $\ell_1$, otherwise $\ell_3$
- Synthetic 3: If $x_{11} < 0$ we use $\ell_2$, otherwise $\ell_3$

The logits have been adapted from the originals in (Yoon et al., 2019) to produce probabilities closer to 0 or 1. This is so all the models have stronger purely predictive performance. The train set is size 60,000, and the validation and test sets are both size 10,000. AUROC is used as the evaluation metric.

**Cube.** The Cube dataset is a synthetic dataset that is regularly used to evaluate Active Feature Acquisition methods (Rückstieß et al., 2013; Shim et al., 2018; Zannone et al., 2019). We specifically use the version where all features are normally distributed (Zannone et al., 2019). There are twenty continuous features, where different features are relevant for different classes. All features are drawn from a normal distribution with mean 0.5 and standard deviation 0.3, except for the following cases:

- Class 1: Features 1, 2, 3 have mean $[0, 0, 0]$ and diagonal standard deviation $[0.1, 0.1, 0.1]$.
- Class 2: Features 2, 3, 4 have mean $[1, 0, 0]$ and diagonal standard deviation $[0.1, 0.1, 0.1]$.
- Class 3: Features 3, 4, 5 have mean $[0, 1, 0]$ and diagonal standard deviation $[0.1, 0.1, 0.1]$.
- Class 4: Features 4, 5, 6 have mean $[1, 1, 0]$ and diagonal standard deviation $[0.1, 0.1, 0.1]$.
- Class 5: Features 5, 6, 7 have mean $[0, 0, 1]$ and diagonal standard deviation $[0.1, 0.1, 0.1]$.
- Class 6: Features 6, 7, 8 have mean $[1, 0, 1]$ and diagonal standard deviation $[0.1, 0.1, 0.1]$.
- Class 7: Features 7, 8, 9 have mean $[0, 1, 1]$ and diagonal standard deviation $[0.1, 0.1, 0.1]$.
- Class 8: Features 8, 9, 10 have mean $[1, 1, 1]$ and diagonal standard deviation $[0.1, 0.1, 0.1]$.

We use a train set with size 60,000, and the validation and test sets are both size 10,000. Accuracy is the evaluation metric.

**Bank Marketing.** The Bank Marketing dataset (Moro et al., 2014) can be found at: `https://archive.ics.uci.edu/dataset/222/bank+marketing`. The data is taken from a marketing campaign conducted by a Portuguese bank. The task is binary classification, where the label indicates whether a client subscribed to a term deposit at the bank. The features are both the client's information and information about the calls. There are 15 features in total (after combining the month and day of the call into one feature), 7 are continuous and 8 are categorical. A full list of features can be found at the dataset origin. We use an 80:10:10 split, giving train, validation, and test sizes of 36,168, 4,521, and 4,522. The evaluation metric is AUROC.

**California Housing.** The California Housing dataset is obtained through Scikit-Learn (Pedregosa et al., 2011) `https://scikit-learn.org/stable/modules/generated/sklearn.datasets.fetch_california_housing.html`. The labels are median house prices in California districts expressed in 100,000 dollars. There are 8 continuous features that can be found at the above URL. To convert this to a classification task, we discretize the labels into 4 equally sized bins. We use an 80:10:10 split, giving train, validation, and test sizes of 16,512, 2,064, and 2,064. The evaluation metric is accuracy.

**MiniBooNE.** MiniBooNE is an experiment at Fermilab designed to detect neutrino oscillations, namely muon-neutrinos into electron-neutrinos (Roe et al., 2005; Roe, 2010). The data was obtained from `https://archive.ics.uci.edu/dataset/199/miniboone+particle+identification`. The task is binary classification, distinguishing electron-neutrino events from background events. The dataset does not have balanced classes. In early experiments, we saw that this meant all models were highly predictive with very few features, and results were indistinguishable. To make the task *harder* for the models, we enforced balance by reducing the number of background events at random to match the number of signal events. We also used a subset of the features selected using STG (Yamada et al., 2020), as a preprocessing step. The selected features were [2, 3, 6, 14, 15, 17, 20, 21, 22, 23, 25, 26, 29, 34, 39, 40, 41, 42, 43, 44]. The train set is size 56,499, and the validation and test sets are both size 10,000. The evaluation metric is AUROC.

**MNIST and Fashion MNIST.** MNIST and Fashion MNIST are image classification datasets with 10 classes, consisting of images of handwritten digits and items of clothing, respectively. We preprocess by selecting a subset of 20 pixels each for computational reasons - an acquisition trajectory with 784 features, where the majority are redundant, will be very slow, especially for methods such as EDDI and VAE, where the whole acquisition is $\mathcal{O}(d^2)$. To do this we use STG (Yamada et al., 2020) a deep learning method for feature selection. After flattening the images to vectors, the features found by STG were:

- MNIST: [153, 154, 210, 211, 243, 269, 271, 295, 327, 348, 350, 375, 405, 409, 427, 430, 461, 514, 543, 655]

- Fashion MNIST: [10, 38, 121, 146, 202, 246, 248, 341, 343, 362, 406, 434, 454, 490, 546, 574, 580, 602, 742, 770]

For both datasets, we split the provided train set into a train set with size 50,000 and validation set with size 10,000. We use the provided test sets, each with size 10,000. The evaluation metric is accuracy.

**METABRIC.** The Molecular Taxonomy of Breast Cancer International Consortium (METABRIC) database consists of clinical and genetic data for 1,980 breast cancer subjects (Curtis et al., 2012; Pereira et al., 2016). The data was accessed at `https://www.kaggle.com/datasets/raghadalharbi/breast-cancer-gene-expression-profiles-metabric`. We construct a classification task, predicting the PAM50 status using gene expressions as features. There are six classes:

1. Luminal A
2. Luminal B
3. HER2 Enriched
4. Claudin Low
5. Basal Low
6. Normal

As with the other high-dimensional datasets, we used STG to select a subset of twelve continuous gene expressions given by:

1. CCNB1
2. CDK1
3. E2F2
4. E2F7
5. STAT5B
6. Notch 1
7. RBPJ
8. Bcl-2
9. eGFR
10. ERBB2
11. ERBB3
12. ABCB1

We use an 80:10:10 split, resulting in train, validation, and test sizes of 1,518, 189, and 191. The evaluation metric is accuracy.

**TCGA.** The Cancer Genome Atlas (TCGA) consists of genetic data for over 11,000 cancer patients (Weinstein et al., 2013). The data was accessed at `https://www.cancer.gov/ccg/research/genome-sequencing/tcga`. We construct the classification task of predicting the location of the tumor based on DNA methylation data. We use 17 locations as the classes:

1. Breast
2. Lung
3. Kidney
4. Brain
5. Ovary
6. Endometrium
7. Head and Neck
8. Central Nervous System
9. Thyroid
10. Prostate
11. Colon
12. Stomach
13. Bladder
14. Liver
15. Cervix
16. Bone Marrow
17. Pancreas

As the first step of dimensionality reduction, we removed features with more than 15% missingness. Following this, we used STG to select 21 features:

1. C7orf51
2. DEF6
3. DNASE1L3
4. EFS

| | | | |
|---|---|---|---|
| 5. FOXE1 | 6. GPR81 | 7. GRIA2 | 8. GSDMC |
| 9. HOXA9 | 10. KAAG1 | 11. KLF5 | 12. LOC283392 |
| 13. LTBR | 14. LYPLAL1 | 15. PON3 | 16. POU3F3 |
| 17. SERPINB1 | 18. ST6GAL1 | 19. TMEM106A | 20. ZNF583 |
| 21. ZNF790 | | | |

We then removed subjects with more than 10% missing features and used an 80:10:10 split. This gave train, validation, and test sizes of 6,327, 790, and 792. The evaluation metric is accuracy.

# I. Model Details and Implementations

All models were implemented using PyTorch (Paszke et al., 2017); code is available at `https://github.com/a-norcliffe/SEFA`.

## I.1. General Model Details

Here we provide details that tend to be shared across models. We explicitly state if a model does not follow the descriptions below and provide model-specific details in the next subsection.

**Input Representation.** Continuous features are represented as $[\mathbf{x} \odot \mathbf{m}, \mathbf{m}]$. Categorical features use a one-hot encoding, where we include an additional class to indicate a missing feature. Continuous and categorical representations are then concatenated as input to the main model.

**Deep Networks.** All deep networks are given by MLPs with ReLU activation followed by Batch Normalization (Ioffe & Szegedy, 2015). All hidden layers in a given network are the same width, which is a hyperparameter that can be tuned as well as the number of hidden layers. The exception to this is the Opportunistic RL model, where we replace Batch Normalization with dropout with $0.5$ probability in accordance with the method's implementation (Kachuee et al., 2019a).

**Acquiring Features.** After a method calculates its acquisition objective $R_{\text{Method}}(\mathbf{x}_O, i)$, this is multiplied by $(1 - m_i)$ so we do not acquire features we have already observed. We also multiply by $m_{\text{Test Data},i}$ so we do not acquire features that are not available in the test data. This would not apply at deployment where we have the ability to measure all features.

## I.2. Model Specific Details

Here we include any key details that are specific to given models, such as hyperparameter names and roles. We highly recommend seeing each method's paper for full details of each model. Unless otherwise stated, each method follows the general rules stated previously.

**Fixed MLP.** The Fixed MLP uses a simple MLP structure as described above. It is trained for 120 epochs. We prevent overfitting during training by choosing the iteration with the best validation accuracy/AUROC. The greedy fixed order is found after training by masking out all features and calculating the evaluation metric on the train set for each feature individually. The best feature is chosen and is unmasked for the model. The procedure is repeated with the best feature being kept to find the second best feature. This is repeated until all features have been placed in a fixed greedy order.

**GDFS.** GDFS (Covert et al., 2023) has two separate networks, one for prediction, one for scoring features. Both have a softmax final activation to give a probability distribution over the label and a positive score for each feature. Our implementation follows the original. We use the same hidden width and number of hidden layers for both networks. The Boolean "Share Parameters" hyperparameter says whether to share half the hidden layers between the two networks, this is presented in the paper as a possible way to increase performance, we treat it as a hyperparameter. We carry out pretraining on the predictor network for 80 epochs, we then carry out main training on both networks. This is done using a geometric temperature progression of $T \times [1.00, 0.56, 0.32, 0.18, 0.1]$, where the initial temperature $T$ is a hyperparameter. Main training is carried out for 15 epochs for each temperature in the progression, please see the original paper for full details. Main training consists of sampling feature acquisitions and training the scoring network to choose features with the best greedy prediction from the predictor network.

**DIME.** DIME (Gadgil et al., 2024) uses two separate networks, one for prediction and one for predicting the CMI of features with the label. The information network is used to score each feature. The information network limits the output

to a minimum of zero and maximum of the entropy of the current predictions $H(Y|\mathbf{x}_S)$. The majority of the DIME implementation follows the GDFS implementation above. Instead of a temperature progression, we use an $\epsilon$ progression during main training following the original paper. This is given by $\epsilon$ Initial$\times$[1.0, 0.25, 0.05, 0.005]. This gives the probability of choosing a feature uniformly at random compared to the best feature predicted by the information network. Main training is also done for 15 epochs for each $\epsilon$ value, the information network is trained to predict the change in loss when a given feature is acquired.

**Opportunistic RL.** Opportunistic RL (Kachuee et al., 2019a) is a Deep Q learning method, where the reward is given by the $l_1$ norm of the change in prediction distribution after an acquisition. The target network is updated compared to the main network with a rate of 0.001 as suggested. Predictions are made by using dropout to provide different network parameters at test time with 50 samples taken and averaged as suggested. The P and Q networks share representations as described in the paper. The model is trained for 20000 episodes with evaluation every 100 episodes. For the first 2000 episodes only the predictor network is trained, using uniformly random actions. Following this, the probability of a random action decays by $0.1^{\frac{1}{20000}}$ every episode to a minimum of 0.1. After 10000 episodes and for every 2000 episodes after that, the learning rate decays by a factor of 0.2. This is all in line with the original implementation. The only change is that in each episode we do not consider individual samples from the dataset (it is not an online stream of data); instead, we train using a batch of samples each episode. This *improves* the training, *improving* Opportunistic RL compared to its original online setting.

**VAE.** The VAE method is a vanilla generative modeling approach to the AFA problem to analyze the viability of generative models. We use a Variational Auto-Encoder (Kingma, Diederik P and Welling, Max, 2014) to model the distribution of the features. We train with the standard ELBO. The encoder and decoder are separate networks with separate widths and numbers of hidden layers. We then train a separate predictor that uses a standard MLP. Features are scored by taking samples of the unknown features conditioned on the observed ones (we use 50 samples). These samples go through the predictor to give an estimated label distribution. The mutual information is then estimated with $I(X_i; Y|\mathbf{x}_S) = \mathbb{E}_{p(x_i|\mathbf{x}_S)}[D_{\text{KL}}(p(Y|x_i, \mathbf{x}_S)||p(Y|\mathbf{x}_S))]$. We train for 120 epochs. We prevent overfitting during training by choosing the iteration with the best validation ELBO.

**EDDI.** EDDI (Ma et al., 2019) is an advanced generative modeling method for AFA. The encoder is a Partial VAE. We encode the label in the same way as a categorical feature. We do not include a separate predictor, instead we follow the original paper to make predictions: features are encoded to a latent distribution and samples are decoded to $y$. The absence of a dedicated predictor negatively impacts the results for EDDI. The decoder follows the same structure as for the VAE. We train for 400 epochs, to prevent overfitting we choose the iteration with the best validation ELBO. Features are scored based on a sampled KL divergence calculated in the latent space as described in the original paper, we use 50 samples.

**ACFlow.** ACFlow is another advanced generative modeling method that is designed to handle arbitrary conditional likelihoods. It works by using a flow based model to maximize the conditional likelihood of $p((\mathbf{x}, y)_U|(\mathbf{x}, y)_O)$, such that arbitrary subsets of the features and label can be used as the condition for other arbitrary sets. We follow the publicly available implementation, training for 120 epochs.

**GSMRL.** GSMRL is an RL method that uses information from a generative surrogate model as additional information to a policy network. As suggested in the original paper, the policy is trained with PPO (Schulman et al., 2017), and a trained ACFlow model is used as the surrogate model. We train for 300 episodes, each episode consisting of 8 epochs.

**SEFA.** Our method, as described in the main paper, encodes each feature separately to a normal distribution. The structure of the individual encoders depends on whether the feature is continuous or categorical:

- **Continuous Features.** Continuous features first undergo a copula transformation $\tilde{x}_i = \Phi^{-1}(F_i(x_i))$, where $F_i$ is the empirical CDF of the feature and $\Phi$ is the standard normal CDF. This transformation is *specific to stochastic encoders* that regularize $I(X_S; Z)$, since it enforces a symmetry associated with the mutual information and also encourages sparse disentangled representations (Wieczorek et al., 2018). We give $[m_i\tilde{x}_i, m_i]$ to that feature's encoder, which is given by the MLP described previously. The MLP ends with Batch Normalization which we found sped up training.

- **Categorical Features.** Categorical features replace the MLP and copula transform by simply having a learnable embedding matrix (where we include an additional category representing a missing feature).

To train, we subsample the features first and use the predictive loss as described in Section 5.1. We use 100 latent samples for training and train for 120 epochs. We also provide pseudo-code for the loss calculation for batch size 1 in Algorithm 1.

To calculate the acquisition objective, we use 200 latent samples and the objective given by (2). We provide pseudo-code for scoring a single feature in Algorithm 2 (the actual implementation is able to score features in parallel on a batch of data).

---

**Algorithm 1** Loss calculation on batch size 1.

---

**Input:** features $\mathbf{x}$, data mask $\mathbf{m}$, label $y$
$\mu = []$
$\sigma = []$
$p_{\text{Removal}} \sim \mathcal{U}(0,1)$
**for** $f = 1$ **to** $d$ **do**
  $u \sim \mathcal{U}(0,1)$
  $\mu_f, \sigma_f = f^e_{\theta_f}(x_f, m_f \times \mathcal{I}[u > p_{\text{Removal}}])$
  $\mu = [\mu, \mu_f]$
  $\sigma = [\sigma, \sigma_f]$
**end for**
$L_R = D_{\text{KL}}(\mathcal{N}(\mu, \sigma^2) \| \mathcal{N}(0,1))$
$p(Y|\mathbf{x}_O) = 0$
**for** sample $= 1$ **to** $N$ **do**
  $\mathbf{z} \sim \mathcal{N}(\mu, \sigma^2)$
  $p(Y|\mathbf{x}_O) = p(Y|\mathbf{x}_O) + \frac{1}{N} f^p_\phi(\mathbf{z})$
**end for**
$L_{\text{NLL}} = -\log(p(Y = y|\mathbf{x}_O))$
**Return:** $L_{\text{NLL}} + \beta L_R$

---

**Algorithm 2** Calculating the acquisition score for feature $i$.

---

**Input:** features $\mathbf{x}$, observation mask $\mathbf{m}$
$\mu = []$
$\sigma = []$
**for** $f = 1$ **to** $d$ **do**
  $\mu_f, \sigma_f = f^e_{\theta_f}(x_f, m_f)$
  $\mu = [\mu, \mu_f]$
  $\sigma = [\sigma, \sigma_f]$
**end for**
$p(Y|\mathbf{x}_O) = 0$
**for** sample $= 1$ **to** $N_1$ **do**
  $\mathbf{z} \sim \mathcal{N}(\mu, \sigma^2)$
  $p(Y|\mathbf{x}_O) = p(Y|\mathbf{x}_O) + \frac{1}{N_1} f^p_\phi(\mathbf{z})$
**end for**
Score $= 0$
**for** $y = 1$ **to** $C$ **do**
  **for** sample $= 1$ **to** $N_2$ **do**
    $\mathbf{z} \sim \mathcal{N}(\mu, \sigma^2)$
    $\mathbf{g} = \nabla_{\mathbf{z}}(f^p_\phi(\mathbf{z})_y)$
    Score $=$ Score $+ p(Y = y|\mathbf{x}_O)\frac{1}{N_2}\frac{\|\mathbf{g}_{\mathcal{G}_i}\|_2}{\sum_j \|\mathbf{g}_{\mathcal{G}_j}\|_2}$
  **end for**
**end for**
**Return:** Score

---

## J. Model Runtimes

There are two places to consider runtime: training and acquisition. In Table 7 we provide the computational complexities of each method with respect to the number of features $d$.

As RL, DIME, and GDFS train by simulating acquisition, training time scales linearly with the number of features. Generative models (and SEFA) are constant to train since they only train to predict well. However, during inference, RL, DIME, and GDFS only require one forward pass of their policy/CMI network, whereas EDDI and VAE must individually score every feature. SEFA takes gradients of the predicted class outputs, so the runtime is linear in the number of classes, which is typically far fewer than the number of features. The main takeaway is that SEFA scales better than half the methods at training time, better than the other half during acquisition (assuming fewer labels than features), and never the worst.

*Table 7.* Computational complexities for the models that actively acquire features, treating a forward pass of a neural network and taking the maximum among $d$ values as approximately constant compared to calculating $R(\mathbf{x}_O, i)$.

| Model | Computational Complexities | | Single Acquisition Times (s) | | | Training Time (s) |
|---|---|---|---|---|---|---|
| | 1 Acquisition Step | 1 Training Step | Syn 1 | MiniBooNE | MNIST | Syn 1 |
| ACFlow | $\mathcal{O}(d)$ | $\mathcal{O}(1)$ | $1.483 \pm 0.003$ | $2.208 \pm 0.002$ | $21.318 \pm 0.191$ | $376.885 \pm 1.871$ |
| DIME | $\mathcal{O}(1)$ | $\mathcal{O}(d)$ | $0.017 \pm 0.000$ | $0.021 \pm 0.000$ | $0.024 \pm 0.000$ | $575.283 \pm 1.332$ |
| EDDI | $\mathcal{O}(d)$ | $\mathcal{O}(1)$ | $2.871 \pm 0.059$ | $9.039 \pm 0.245$ | $12.360 \pm 0.029$ | $273.358 \pm 1.060$ |
| GDFS | $\mathcal{O}(1)$ | $\mathcal{O}(d)$ | $0.018 \pm 0.000$ | $0.020 \pm 0.000$ | $0.024 \pm 0.000$ | $277.651 \pm 0.716$ |
| GSMRL | $\mathcal{O}(1)$ | $\mathcal{O}(d)$ | $0.047 \pm 0.000$ | $0.045 \pm 0.000$ | $0.097 \pm 0.001$ | $1010.650 \pm 3.186$ |
| Opportunistic RL | $\mathcal{O}(1)$ | $\mathcal{O}(d)$ | $0.029 \pm 0.001$ | $0.034 \pm 0.001$ | $0.030 \pm 0.000$ | $716.517 \pm 2.962$ |
| VAE | $\mathcal{O}(d)$ | $\mathcal{O}(1)$ | $0.153 \pm 0.000$ | $0.291 \pm 0.001$ | $0.452 \pm 0.007$ | $126.774 \pm 1.403$ |
| SEFA (ours) | $\mathcal{O}(|y|)$ | $\mathcal{O}(1)$ | $0.246 \pm 0.001$ | $0.314 \pm 0.002$ | $1.498 \pm 0.002$ | $297.061 \pm 1.701$ |

In Table 7, we also include the time per acquisition step on a subset of the datasets. As expected, the acquisition time for

SEFA does not increase significantly with the number of features (Syn 1 to MiniBooNE), but does as the number of classes increases (MiniBooNE to MNIST). The generative models scale the worst, and models with policy networks are the fastest to make acquisitions. For completeness, we also include the total training times for Syn 1. These results should be treated carefully, since this depends on the number of epochs, different methods converge at different rates. However, the pattern is as expected, the models that train a policy network by simulating acquisition are slower to train than the generative models and SEFA. The main takeaway from the recorded wall-clock times is the same as for the computational complexities, SEFA scales better than half the methods at training time, better than the other half at acquisition time, and never the worst.

## K. Experimental Details

All experiments were run on an Nvidia Quadro RTX 8000 GPU. The specifications can be found at https://www.nvidia.com/content/dam/en-zz/Solutions/design-visualization/ quadro-product-literature/quadro-rtx-8000-us-nvidia-946977-r1-web.pdf. All experiments were repeated five times over parameter initializations to obtain means and standard error estimates.

**Training.** We train all models using the Adam optimizer (Kingma & Ba, 2015), the learning rate and batch size are hyperparameters that are tuned using a validation set. All methods (except for Opportunistic RL which uses its original implementation) use a learning rate scheduler that multiplies the learning rate by 0.2 when there have been a set number of epochs without validation metric improvement - the patience, which is also tuned.

We prevent overfitting during training by tracking a validation metric every epoch and using the model parameters that produce the best value. The validation metric we choose (unless explicitly stated for a given model) is the area under the acquisition curve. Starting from zero features, we acquire features individually, calculating the accuracy/AUROC at each acquisition, and then the validation metric is the area under the acquisition curve divided by the total number of features.

**Hyperparameter Tuning.** For every model, initial hyperparameter tuning was conducted by finding ranges for each hyperparameter that produced strong acquisition performance on the synthetic datasets. Following this, for each model, we generated 9 hyperparameter configurations using the ranges. For each method, we test each configuration three times producing a mean value for the area under the acquisition curve. The configuration with the highest mean value is separately trained five times in the main experiments. The nine configurations for each method are provided in Tables 8, 9, 10, 11, 12, 13, 14, 15 and 16. We give the selected hyperparameter configurations for each dataset in Table 17.

*Table 8.* Hyperparameter configurations for ACFlow.

| Hyperparameter | 1 | 2 | 3 | 4 | 5 | 6 | 7 | 8 | 9 |
|---|---|---|---|---|---|---|---|---|---|
| Prior Hidden Width | 100 | 200 | 200 | 150 | 100 | 100 | 120 | 60 | 60 |
| No. Hidden Prior Layers | 1 | 2 | 2 | 1 | 1 | 2 | 1 | 2 | 2 |
| No. Flow Modules | 3 | 3 | 5 | 5 | 3 | 5 | 4 | 5 | 5 |
| Flow Module Hidden Width | 100 | 100 | 100 | 150 | 100 | 100 | 120 | 60 | 60 |
| No. Hidden Flow Module Layers | 1 | 1 | 1 | 1 | 1 | 2 | 1 | 2 | 2 |
| $\lambda_{\text{NLL}}$ | 0.1 | 0.1 | 0.05 | 0.05 | 0.01 | 0.3 | 0.1 | 0.1 | 0.1 |
| Learning Rate | 0.001 | 0.001 | 0.001 | 0.001 | 0.0005 | 0.001 | 0.0005 | 0.0005 | 0.001 |
| Batchsize | 128 | 128 | 256 | 256 | 128 | 256 | 128 | 128 | 256 |
| Patience | 5 | 5 | 5 | 5 | 5 | 5 | 3 | 3 | 3 |

*Table 9.* Hyperparameter configurations for DIME.

| Hyperparameter | 1 | 2 | 3 | 4 | 5 | 6 | 7 | 8 | 9 |
|---|---|---|---|---|---|---|---|---|---|
| Hidden Width | 200 | 200 | 200 | 200 | 100 | 200 | 100 | 100 | 100 |
| No. Hidden Layers | 2 | 2 | 2 | 2 | 2 | 2 | 1 | 3 | 1 |
| Share Parameters | False | False | False | True | True | True | False | False | True |
| Pretraining Learning Rate | 0.001 | 0.001 | 0.001 | 0.001 | 0.001 | 0.001 | 0.001 | 0.001 | 0.001 |
| Main Training Learning Rate | 0.001 | 0.001 | 0.001 | 0.001 | 0.001 | 0.001 | 0.001 | 0.001 | 0.0001 |
| Batch Size | 128 | 128 | 128 | 128 | 128 | 128 | 512 | 256 | 512 |
| Patience | 5 | 5 | 5 | 5 | 2 | 5 | 5 | 3 | 5 |
| $\epsilon$ Initial | 0.4 | 0.2 | 0.1 | 0.4 | 0.2 | 0.1 | 0.4 | 0.2 | 0.1 |

*Table 10.* Hyperparameter configurations for EDDI.

| Hyperparameter | 1 | 2 | 3 | 4 | 5 | 6 | 7 | 8 | 9 |
|---|---|---|---|---|---|---|---|---|---|
| C Dim | 200 | 200 | 50 | 100 | 20 | 80 | 250 | 100 | 60 |
| Latent Width | 200 | 200 | 100 | 50 | 20 | 80 | 250 | 40 | 60 |
| No. Hidden Decoder Layers | 2 | 2 | 2 | 2 | 2 | 1 | 2 | 2 | 1 |
| Decoder Hidden Width | 200 | 200 | 200 | 200 | 200 | 100 | 100 | 75 | 200 |
| No. Hidden Encoder Layers | 2 | 2 | 2 | 2 | 2 | 2 | 3 | 2 | 3 |
| Encoder Hidden Width | 200 | 200 | 200 | 200 | 200 | 100 | 100 | 75 | 200 |
| Learning Rate | 0.001 | 0.001 | 0.001 | 0.001 | 0.001 | 0.001 | 0.001 | 0.001 | 0.001 |
| Batch Size | 128 | 512 | 128 | 256 | 512 | 128 | 128 | 256 | 512 |
| $\sigma$ Decoder | 0.2 | 1.0 | 0.2 | 0.2 | 0.2 | 0.2 | 0.2 | 0.2 | 0.2 |
| Patience | 5 | 5 | 5 | 5 | 5 | 5 | 5 | 5 | 5 |

*Table 11.* Hyperparameter configurations for Fixed MLP.

| Hyperparameter | 1 | 2 | 3 | 4 | 5 | 6 | 7 | 8 | 9 |
|---|---|---|---|---|---|---|---|---|---|
| Hidden Width | 200 | 100 | 200 | 100 | 300 | 100 | 250 | 50 | 120 |
| No. Hidden Layers | 2 | 2 | 1 | 1 | 2 | 2 | 3 | 2 | 2 |
| Learning Rate | 0.001 | 0.001 | 0.001 | 0.001 | 0.001 | 0.001 | 0.001 | 0.001 | 0.0005 |
| Batch Size | 128 | 128 | 128 | 128 | 256 | 128 | 256 | 64 | 128 |
| Patience | 5 | 5 | 5 | 5 | 5 | 2 | 10 | 5 | 5 |

*Table 12.* Hyperparameter configurations for GDFS.

| Hyperparameter | 1 | 2 | 3 | 4 | 5 | 6 | 7 | 8 | 9 |
|---|---|---|---|---|---|---|---|---|---|
| Hidden Width | 200 | 200 | 200 | 200 | 200 | 200 | 100 | 200 | 200 |
| No. Hidden Layers | 2 | 2 | 2 | 2 | 2 | 2 | 1 | 2 | 2 |
| Share Parameters | False | False | False | True | True | True | True | False | True |
| Pretraining Learning Rate | 0.001 | 0.001 | 0.001 | 0.001 | 0.001 | 0.001 | 0.001 | 0.001 | 0.001 |
| Main Training Learning Rate | 0.001 | 0.001 | 0.001 | 0.001 | 0.001 | 0.001 | 0.001 | 0.001 | 0.001 |
| Batch Size | 128 | 128 | 128 | 128 | 128 | 128 | 512 | 512 | 512 |
| Patience | 2 | 2 | 2 | 2 | 2 | 2 | 2 | 2 | 2 |
| Temp Initial | 2.0 | 1.0 | 0.1 | 2.0 | 1.0 | 0.1 | 2.0 | 1.0 | 0.1 |

*Table 13.* Hyperparameter configurations for GSMRL.

| Hyperparameter | 1 | 2 | 3 | 4 | 5 | 6 | 7 | 8 | 9 |
|---|---|---|---|---|---|---|---|---|---|
| Hidden Width | 100 | 100 | 200 | 100 | 100 | 150 | 75 | 200 | 100 |
| No. Hidden Actor Layers | 2 | 2 | 1 | 2 | 2 | 3 | 2 | 1 | 2 |
| No. Hidden Critic Layers | 2 | 2 | 1 | 2 | 2 | 3 | 2 | 1 | 2 |
| Discount Factor $\gamma$ | 0.95 | 0.95 | 0.9 | 0.99 | 0.99 | 0.95 | 0.9 | 0.99 | 0.95 |
| $\lambda_{\text{GAE}}$ | 0.95 | 0.95 | 0.9 | 0.95 | 0.9 | 0.95 | 0.8 | 0.8 | 0.95 |
| $\lambda_{\text{Entropy}}$ | 0.01 | 0.01 | 0.0 | 0.005 | 0.0 | 0.0 | 0.01 | 0.01 | 0.1 |
| Gradient Clip Max Norm | 1.0 | 1.0 | 0.5 | 1.0 | 1.0 | 1.0 | 1.0 | 1.0 | 1.0 |
| No. Epochs per Episode | 5 | 5 | 8 | 5 | 5 | 5 | 10 | 10 | 8 |
| Rollout Batch Size | 256 | 256 | 256 | 512 | 512 | 512 | 300 | 256 | 256 |
| Optimization Batch Size | 128 | 128 | 128 | 128 | 256 | 256 | 100 | 50 | 50 |
| Use Surrogate Reward | True | False | False | True | True | False | True | False | True |
| No. Auxiliary Samples | 5 | 5 | 3 | 5 | 5 | 10 | 5 | 5 | 3 |
| Learning Rate | 0.001 | 0.001 | 0.001 | 0.001 | 0.0005 | 0.0001 | 0.0001 | 0.0005 | 0.001 |
| Patience | 5 | 5 | 5 | 2 | 2 | 3 | 5 | 3 | 2 |

*Table 14.* Hyperparameter configurations for Opportunistic RL.

| Hyperparameter | 1 | 2 | 3 | 4 | 5 | 6 | 7 | 8 | 9 |
|---|---|---|---|---|---|---|---|---|---|
| Hidden Width | 200 | 200 | 200 | 100 | 200 | 200 | 100 | 200 | 100 |
| No. Hidden Layers | 2 | 2 | 2 | 2 | 2 | 2 | 1 | 1 | 1 |
| Discount Factor $\gamma$ | 0.5 | 0.75 | 0.25 | 0.5 | 0.75 | 0.25 | 0.5 | 0.75 | 0.25 |
| Learning Rate | 0.001 | 0.001 | 0.001 | 0.001 | 0.001 | 0.0001 | 0.001 | 0.0001 | 0.001 |
| Batch Size | 128 | 128 | 128 | 256 | 256 | 128 | 256 | 256 | 128 |

**Stochastic Encodings for Active Feature Acquisition**

*Table 15.* Hyperparameter configurations for VAE.

| Hyperparameter | 1 | 2 | 3 | 4 | 5 | 6 | 7 | 8 | 9 |
|---|---|---|---|---|---|---|---|---|---|
| Latent Width | 30 | 10 | 50 | 30 | 50 | 10 | 30 | 40 | 20 |
| No. Hidden Decoder Layers | 2 | 2 | 2 | 1 | 2 | 2 | 2 | 2 | 3 |
| Decoder Hidden Width | 100 | 200 | 150 | 200 | 200 | 100 | 100 | 200 | 250 |
| No. Hidden Encoder Layers | 2 | 2 | 2 | 1 | 1 | 2 | 2 | 2 | 3 |
| Encoder Hidden Width | 100 | 200 | 150 | 200 | 150 | 100 | 100 | 200 | 250 |
| No. Hidden Predictor Layers | 2 | 2 | 2 | 1 | 2 | 2 | 2 | 2 | 2 |
| Predictor Hidden Width | 100 | 100 | 200 | 200 | 200 | 100 | 100 | 200 | 100 |
| Learning Rate | 0.001 | 0.001 | 0.001 | 0.001 | 0.001 | 0.001 | 0.0005 | 0.001 | 0.001 |
| Batch Size | 128 | 128 | 128 | 128 | 256 | 512 | 64 | 128 | 512 |
| $\sigma$ Decoder | 0.2 | 0.2 | 0.2 | 0.2 | 0.2 | 1.0 | 0.2 | 0.2 | 0.2 |
| Patience | 5 | 5 | 5 | 5 | 5 | 5 | 5 | 3 | 5 |

*Table 16.* Hyperparameter configurations for SEFA.

| Hyperparameter | 1 | 2 | 3 | 4 | 5 | 6 | 7 | 8 | 9 |
|---|---|---|---|---|---|---|---|---|---|
| Latent Components per Feature | 4 | 4 | 4 | 6 | 4 | 8 | 4 | 6 | 8 |
| No. Hidden Predictor Layers | 2 | 2 | 2 | 2 | 2 | 1 | 2 | 3 | 2 |
| Predictor Hidden Width | 100 | 250 | 100 | 150 | 250 | 250 | 180 | 250 | 250 |
| No. Hidden Encoder Layers | 2 | 2 | 2 | 2 | 2 | 1 | 2 | 3 | 2 |
| Encoder Hidden Width | 20 | 150 | 20 | 50 | 150 | 100 | 40 | 100 | 100 |
| $\beta$ | 0.0005 | 0.001 | 0.001 | 0.0005 | 0.005 | 0.0001 | 0.0008 | 0.001 | 0.005 |
| Learning Rate | 0.001 | 0.0005 | 0.001 | 0.001 | 0.0003 | 0.001 | 0.0005 | 0.0005 | 0.0005 |
| Batch Size | 128 | 128 | 128 | 128 | 128 | 256 | 128 | 256 | 128 |
| Patience | 5 | 5 | 5 | 5 | 5 | 5 | 8 | 5 | 5 |

*Table 17.* Selected hyperparameter configurations for each dataset.

| Dataset | ACFlow | DIME | EDDI | Fixed MLP | GDFS | GSMRL | Opportunistic RL | VAE | SEFA |
|---|---|---|---|---|---|---|---|---|---|
| Syn 1 | 7 | 5 | 3 | 7 | 9 | 9 | 1 | 1 | 4 |
| Syn 2 | 1 | 6 | 5 | 7 | 6 | 9 | 1 | 2 | 1 |
| Syn 3 | 7 | 4 | 3 | 7 | 6 | 7 | 5 | 8 | 6 |
| Cube | 5 | 4 | 4 | 3 | 6 | 7 | 3 | 2 | 5 |
| Bank Marketing | 1 | 4 | 8 | 7 | 3 | 3 | 4 | 9 | 4 |
| California Housing | 1 | 6 | 7 | 7 | 5 | 8 | 3 | 3 | 7 |
| MiniBooNE | 5 | 4 | 9 | 7 | 2 | 9 | 6 | 9 | 7 |
| MNIST | 5 | 6 | 7 | 7 | 6 | 7 | 3 | 3 | 8 |
| Fashion MNIST | 5 | 4 | 4 | 7 | 9 | 7 | 3 | 5 | 4 |
| METABRIC | 2 | 5 | 4 | 5 | 2 | 6 | 9 | 4 | 5 |
| TCGA | 1 | 4 | 4 | 1 | 2 | 6 | 6 | 5 | 4 |