# OpenReview forum: "Stochastic Encodings for Active Feature Acquisition"
_ICML.cc/2025/Conference — ICML 2025 poster_

### Official Review · Reviewer_N1oA · 2025-03-13

**Overall Recommendation:** 3

**Summary:**

Considering the training challenges of reinforcement-learning approaches and the myopic shortcomings of conditional mutual information (CMI) strategies, this paper proposed a method called Stochastic Encodings for Active Feature Acquisition (SEFA). By encoding features into a regularized, stochastic latent space, it can better predict future observations; it then scores each unobserved feature using a gradient-based measure of how it would disambiguate among likely classes. This paper tested their method on standard tabular/image data, and cancer gene-expression classification.

**Claims And Evidence:**

Yes.

**Essential References Not Discussed:**

N/A

**Experimental Designs Or Analyses:**

The experimental settings, including datasets, baselines and evaluation metrics are standard.

**Methods And Evaluation Criteria:**

The paper’s benchmark datasets and evaluation setup from synthetic datasets designed to showcase the pitfalls of myopic conditional mutual information to tabular, image, and real-world medical tasks to demonstrate both practical applicability.

**Other Comments Or Suggestions:**

N/A

**Other Strengths And Weaknesses:**

Strengths: The method has been tested across synthetic data, real tabular, image, and cancer genomics datasets.

Weaknesses: The method’s factorized latent encoder implicitly assumes feature independence in the unobserved blocks, however, considering some strong feature interdependencies situations, this method might not work well.

**Questions For Authors:**

Figures like Figure 4 are really hard to read.

**Relation To Broader Scientific Literature:**

SEFA draws inspiration from several established concepts and methods, like RL, CMI. SEFA goes beyond these existing methods by incorporating a stochastic, latent-space perspective that is regularized via an information bottleneck. It introduces a gradient-based sensitivity strategy in which each feature owns a block of the latent representation.

**Theoretical Claims:**

These formal statements (Propositions 4.1 and 4.2) are demonstrated on that indicator toy example. Seems that the authors use it as a focused scenario to prove where purely myopic strategies fall short and how considering possible unobserved values can address that limitation. They do not provide general proofs.

---

> ### Author Rebuttal · Authors · 2025-03-31
>
> Thank you for the positive review, we are grateful for the feedback, we answer your questions below. All changes below will be added.
>
> # Theory
> We agree that the theory provided is specific to the indicator and does not provide general statements about bounds on SEFA's performance. The purpose was not to provide fully general proofs, but insight into why CMI maximization can fail (it does not take into account possible future feature observations).
>
> **Purpose of Proposition 4.2**: Proposition 4.1 gives a necessary condition for optimality, proposition 4.2 shows that it can be sufficient, we include 4.2 because only necessity without showing sufficiency is a weaker result. The connection to SEFA is to use the **idea** of considering possible future observations in the model design (the expectation in the latent space), since this meets a necessary condition for optimality. We verify this empirically in our ablations (Table 1 and Table 3 Deterministic Encoder). We do not use the objective from 4.2 because it is intractable and does not fit into our latent set up.
>
> **Generalizing Beyond the Indicator**: Proposition 4.1 and 4.2 can be generalized beyond the indicator: CMI will be suboptimal where there are features that are jointly informative but individually non-informative. By definition, the CMI of the jointly informative features is zero, so any other features that are individually informative (even if they are very noisy) will be selected first by CMI. In contrast, the objective in 4.2 can capture the effect of jointly informative features, since it considers the possible values of the unobserved features.
>
> # Feature Independence
> There are two reasons why we believe our conclusion overstates this as a limitation:
>
> 1. Whilst the encoders do not model the feature interdependence, this can be accounted for by the predictor network $p\_{\phi}(y|\mathbf{z})$. The complex interdependencies are captured by the layers after the latent space. Existing work uses this principle, for example, $\beta$-VAE [Higgins et al. 2017] has disentangled latent spaces where the complex relationships are handled by the decoder. The key point is that we are not modelling distributions of the features themselves, but of their representation - their effects on the predictor network. The predictor can learn to act differently based on other latent values, for example, using samples of feature 2's latent components differently depending on if latent component 1 is positive or not. Whilst the distribution of feature 2's latent components cannot be affected by observing feature 1, how the latent samples are used by the predictor can be affected. As a result the gradients are affected, and therefore the acquisition scores. This is another reason to run the acquisition in the latent space. The predictor network can learn the complex relationships (provided it trains on enough latent samples), allowing the encoder to have a structure that does not learn the complex feature distributions. Whereas if we ran the acquisition in feature space, the predictor would only be trained on real feature values, and so the generative model would need to be able to learn the complex multivariate feature distributions.
>
> 2. Empirically we see this. The Feature Space Ablation (Table 1 and Table 3), uses a generative model for $p(\mathbf{x}\_{U}|\mathbf{x}\_{O})$, to run the acquisition in feature space and does not perform as well as SEFA that runs in the latent space, despite being able to model the dependencies. Additionally, our datasets have high interdependencies (image and genomics datasets), and SEFA still performs well on these.
>
> We appreciate this is still a modelling restriction, however, we believe that due to the predictor being able to model the complex interdependencies and the ablation results, we have overstated this as a limitation. We will adjust our conclusion to accurately reflect this.
>
> # Figure 4
> When we were making Figure 4, we initially had just the feature numbers on the y-axis. However, the trade-off we made was to put the feature names, providing more immediate information, but making the text smaller. The figure is a vector image so can be zoomed in. We shall add that the figure is best viewed zoomed in to the caption.
>
> As suggested by Reviewer V2wP, without the supporting text it is not immediately clear which features are relevant for certain tumor locations. Therefore, we will edit Figure 4, such that class-wise selections with high frequencies (darker rectangles on the heat map) are highlighted using green bounding boxes, as was done with Syn 1-3. And notable low frequency selections, (where the selection frequency is low, but high for other classes) will be highlighted with black bounding boxes. This way the instance-wise differences are more noticeable, the selections are easy to identify when reading about them in the text, and the figure should be clearer by guiding attention to relevant parts of the figure.

---

### Official Review · Reviewer_Pdip · 2025-03-14

**Overall Recommendation:** 3

**Summary:**

This paper addresses Active Feature Acquisition (AFA), the task of sequentially selecting which features to measure for a specific test instance to improve prediction accuracy while minimizing the number of features acquired. The authors identify limitations in existing approaches: Reinforcement Learning (RL) methods face training difficulties, while Conditional Mutual Information (CMI) maximization makes myopic decisions that don't consider future acquisitions.

The method was evaluated on synthetic datasets with known optimal feature orderings and on real-world datasets including image classification and cancer classification tasks. The authors demonstrate that SEFA consistently outperforms baseline methods, including both RL-based and CMI-based approaches. For the cancer classification task, they also validate that the features selected by SEFA align with biomarkers identified in scientific literature.

**Claims And Evidence:**

Overall, the paper's claims are generally well-supported by evidence, though with some limitations:

## Well-supported claims:

1. **Theoretical limitations of CMI maximization**: The authors provide both theoretical arguments and concrete examples showing why greedy CMI maximization can be sub-optimal for AFA (Section 4).

2. **Performance of SEFA vs. baselines**: The extensive empirical evaluation across multiple datasets consistently shows that SEFA outperforms baseline methods. The authors report means with standard errors across 5 runs, providing statistical confidence in their results.

3. **Ablation studies**: The impact of each SEFA component is demonstrated through comprehensive ablations on synthetic and real datasets, showing that each design element contributes to performance.

4. **Feature relevance on medical datasets**: For TCGA data, the authors cite scientific literature supporting that their model selects biologically relevant features for different cancer types.

## Claims with more limited evidence:

1. **Overcoming RL difficulties**: While the authors cite RL challenges as motivation for SEFA, they don't directly demonstrate how these specific challenges affect performance on their tasks. SEFA does outperform the RL baseline, but the exact connection to the cited RL difficulties isn't experimentally verified.

2. **Non-greedy acquisitions**: The synthetic experiments show SEFA makes better decisions than greedy methods, but it's not entirely clear if this is due to the specific "non-greedy" design or other aspects of the method. The indicator example helps, but more direct experimental validation of this specific claim would strengthen it.

3. **Scalability claims**: While the paper discusses computational complexity in Table 4, there's no empirical evaluation of actual runtime comparisons between methods, which would help verify the practical implications of the theoretical complexities.

4. **Generalization beyond classification**: The method is currently limited to classification tasks, as noted in the limitations section. Claims about the general superiority of the approach should be considered in this context.

The paper's claims are largely substantiated by the evidence presented, with the experimental methodology being thorough and the conclusions reasonably drawn from the results.

**Essential References Not Discussed:**

While the paper provides a thorough literature review, there are a few notable omissions that would help contextualize their contributions:

- Monte Carlo Tree Search (MCTS): The non-greedy acquisition problem shares similarities with planning problems where MCTS has been successful (Lim et al., 2012). MCTS explicitly balances immediate rewards with long-term payoffs through rollouts, conceptually similar to how SEFA samples multiple latent realizations to evaluate acquisition decisions.

- Feature attribution methods: SEFA's gradient-based approach for calculating feature importance shares similarities with feature attribution methods like Integrated Gradients (Sundararajan et al., 2017) and LIME (Ribeiro et al., 2016). These connections could strengthen the theoretical foundation of the gradient-based scoring function.

- Partial observability in sequential decision making: The paper lacks references to Partially Observable Markov Decision Processes (POMDPs) literature

**Experimental Designs Or Analyses:**

The synthetic dataset experiments (Section 6.1) are well-designed to test whether models can learn optimal feature orderings. The use of known optimal paths, multiple runs with error reporting, and clear visualizations provides strong validation of acquisition strategies. This controlled setting effectively demonstrates SEFA's ability to prioritize features optimally.

The real dataset experiments (Section 6.2) appropriately evaluate performance across varied acquisition steps. The diverse dataset selection, consistent protocols, and multiple runs strengthen validity. One limitation is the pre-selection of features for high-dimensional datasets using STG, which might introduce bias if different acquisition methods would naturally prefer different feature subsets.

The cancer classification experiments (Section 6.3) effectively combine quantitative performance with qualitative analysis. The supporting literature citations for selected features and analysis across cancer types validate that the model makes biologically plausible decisions. While not exhaustive, this literature validation adds credibility to the feature selections.

The ablation studies are comprehensive, systematically removing each proposed component while maintaining consistent evaluation protocols across synthetic and real datasets. The accompanying visualizations of acquisition behavior clearly demonstrate each component's impact, strongly supporting the claim that all components are necessary for optimal performance.

The sensitivity analyses and hyperparameter selection procedures follow sound practices, with appropriate exploration of parameter ranges and validation metrics. The reporting of final configurations enhances reproducibility, though more explicit justification for initial hyperparameter ranges would strengthen methodological transparency.

Overall, the experimental designs employ appropriate controls, metrics, and validation procedures. The analyses thoroughly support the paper's claims about SEFA's effectiveness for active feature acquisition.

**Methods And Evaluation Criteria:**

## Proposed Methods

The proposed SEFA method makes sense for the Active Feature Acquisition (AFA) problem and addresses known limitations of existing approaches:

1. **Latent space reasoning**: Using a regularized latent space to simplify decision-making is appropriate for complex feature relationships, particularly since the Information Bottleneck approach helps focus on label-relevant information.

2. **Stochastic encoding**: The approach of using multiple latent samples to consider diverse feature realizations addresses the key issue of myopic decision-making in CMI-based methods.

3. **Probability weighting**: Weighting by predicted class probabilities is a sensible way to focus on distinguishing between likely classes rather than ruling out unlikely ones.

4. **Supervised training**: Avoiding reinforcement learning complexities by using supervised training with a predictive loss is a reasonable design choice given the known difficulties with RL.

The factorization of the latent distribution (each feature responsible for specific latent components) is a pragmatic simplification, though as the authors acknowledge, it limits modeling conditional dependencies between features.

## Evaluation Criteria

The evaluation approach is comprehensive and appropriate:

1. **Dataset selection**: The mix of synthetic datasets (with known optimal strategies), image datasets, tabular datasets, and medical datasets provides good coverage of different application scenarios.

2. **Synthetic datasets**: Using synthetic data with known optimal feature ordering provides clear validation of the method's ability to learn non-myopic acquisition strategies.

3. **Metrics**: Using AUROC for binary classification and accuracy for multi-class classification are standard and appropriate choices. The "average evaluation metric during acquisition" appropriately captures performance across the entire acquisition process.

4. **Baselines**: The comparison against diverse baselines (RL-based, CMI-based, and fixed ordering) provides a thorough evaluation landscape.

5. **Ablation studies**: The detailed ablation experiments effectively isolate the contribution of each component of the method.

6. **Qualitative analysis**: For the TCGA dataset, connecting selected features to literature-supported biomarkers adds biological plausibility to the results.

One potential limitation is the lack of evaluation under different acquisition budget constraints, which would be relevant for resource-constrained applications. Also, while the authors pre-select features for high-dimensional datasets (MNIST, TCGA, etc.) using STG for computational efficiency, this pre-selection might impact the true feature acquisition performance in those domains.

Overall, the methods and evaluation criteria are well-aligned with the AFA problem and provide convincing evidence for the effectiveness of SEFA.

**Other Comments Or Suggestions:**

1. When describing the latent space regularization (Section 5.3), the paper refers to using a variational upper bound but doesn't explicitly state the form of this bound. For clarity, it would be helpful to provide the specific form.

**Other Strengths And Weaknesses:**

## Strengths

**Novel Integration of Techniques**: The paper creatively combines stochastic encoders, latent space regularization, and gradient-based feature scoring into a coherent framework. While each individual component builds on existing ideas, their integration into a unified acquisition approach is original and effective.

**Principled Design Choices**: Each aspect of SEFA addresses specific limitations of prior methods. The stochastic latent space addresses myopic decision-making, probability weighting focuses on distinguishing between likely classes, and the factorized architecture enables tractable feature scoring. These design choices are well-motivated by theoretical insights.

**Strong Empirical Results**: The consistent outperformance across diverse datasets is impressive. Particularly noteworthy is the performance on medical datasets where domain knowledge validates that selected features align with biological mechanisms.

**Pragmatic Balance**: The method strikes a good balance between theoretical principles and practical implementation. By using supervised training instead of reinforcement learning, the authors provide a more accessible approach to AFA that performs better and is easier to implement.

**Clear Writing and Visualizations**: The paper is generally well-written with excellent visualizations. The heat maps showing acquisition trajectories (Figures 2, 4) are particularly effective at conveying complex patterns in feature selection across different scenarios.

## Weaknesses

**Limited Theoretical Guarantees**: While the paper provides examples where SEFA works well, it lacks formal guarantees about its optimality or convergence properties. The theoretical foundation primarily motivates the approach rather than providing rigorous performance bounds.

**Parameter Sensitivity**: The method introduces several hyperparameters (β, number of latent components per feature, number of samples). While ablations show their importance, the paper doesn't provide strong guidance on how to select these parameters for new datasets beyond empirical tuning.

**Computational Overhead**: At inference time, SEFA requires sampling from latent distributions and computing gradients, which may be computationally demanding compared to simpler methods. The paper acknowledges this limitation but doesn't provide empirical runtime comparisons.

**Restricted to Classification**: As noted in the limitations, the method is currently designed for classification tasks. This restricts its applicability to regression problems, which are common in many domains where feature acquisition is relevant (e.g., predictive maintenance, climate modeling).

**Feature Independence Assumption**: The factorized latent space, while computationally convenient, assumes independent encoding of features. This may limit the model's ability to capture complex interactions between features, particularly in datasets with strong feature correlations.

**Limited Discussion of Scalability**: While the paper demonstrates effectiveness on datasets with moderate dimensionality, it's unclear how the approach would scale to problems with thousands or millions of features, which appear in genomics and other high-dimensional domains.

Overall, the paper presents a significant contribution to the AFA literature with strong empirical validation. Its originality lies more in the creative combination of techniques and their application to an important problem than in fundamental theoretical breakthroughs.

**Questions For Authors:**

1. **Computational Overhead Analysis**: How does the inference time of SEFA compare to the baseline methods across different datasets? Since SEFA requires multiple latent samples and gradient calculations during acquisition, it would be helpful to understand the practical trade-off between performance gains and computational costs. A response showing reasonable computational requirements would strengthen the practical applicability of the method.

2. **Adaptation to Varying Acquisition Costs**: Many real-world scenarios have different costs associated with acquiring different features (e.g., some medical tests are more expensive or invasive than others). How would SEFA be adapted to account for variable acquisition costs? A clear adaptation strategy would significantly enhance the method's practical utility in cost-sensitive domains.

3. **Extension to Regression Tasks**: Given that the current probability weighting mechanism is specific to classification, what modifications would be required to extend SEFA to regression tasks? Understanding this extension would clarify the method's generalizability beyond classification problems, which is currently listed as a limitation.

4. **Robustness to Noisy Features**: How does SEFA perform when features contain noise or measurement errors? Since the method relies on gradient information to score features, it's important to understand its sensitivity to noise. Evidence of robustness would strengthen confidence in SEFA's real-world applicability.

5. **Selection of Latent Dimensionality**: How should practitioners determine the number of latent components per feature for new datasets? This hyperparameter seems crucial for balancing model capacity with regularization, and guidance beyond empirical tuning would make the method more accessible to new users.

**Relation To Broader Scientific Literature:**

The paper's critique of CMI-based acquisition relates to broader discussions about greedy vs. non-greedy information acquisition. Chen et al. (2015a) established theoretical bounds for when greedy information maximization is near-optimal, but SEFA demonstrates scenarios where this breaks down. SEFA's factorized latent space approach conceptually relates to Ma et al.'s EDDI (2019), but fundamentally differs by focusing on gradients in latent space rather than estimating CMI.

SEFA addresses the exploration-exploitation tradeoff in active learning without using reinforcement learning. This connects to broader challenges in RL, offering an alternative supervised approach.

SEFA's probability weighting mechanism addresses a known issue in information-theoretic approaches where minimizing entropy can focus on low-probability events. This relates to research on decision-theoretic active learning  and optimal experimental design, where the goal is to directly minimize expected error rather than maximize information gain.
SEFA's approach of encoding features individually before integration relates to research on partial variable-wise encoders in missing data contexts. While prior works like ACFlow (Li et al., 2020) used normalizing flows for this purpose, SEFA provides a simpler yet effective alternative specifically designed for the acquisition task.

**Theoretical Claims:**

1. The proof of Proposition 4.2 doesn't fully explain why considering possible unobserved feature values in the acquisition objective enables optimal feature acquisition more generally beyond the indicator example.

2. The paper doesn't provide formal guarantees about SEFA's optimality, only demonstrating that considering unobserved feature values is necessary for optimality in the indicator example.

3. The connection between the theoretical insights from Propositions 4.1 and 4.2 and the design of SEFA isn't made fully explicit—particularly how the stochastic latent space approach relates to the integral in Proposition 4.2.

---

> ### Author Rebuttal · Authors · 2025-03-31
>
> Thank you for the positive review, we are grateful for the feedback, we answer your questions below. All changes below will be added.
>
> # Scalability
> We have addressed this shared point in our response to Reviewer ABGZ.
>
> # Generalization to Regression
> SEFA can be modified for regression so that the predictor network has two heads, one that predicts the label directly and one that predicts the label as a classification problem (after it has been discretized into equally sized bins). We can use the regression head to predict the label, and the classification head to carry out the feature acquisitions.
>
> # Feature Costs and Budget
> **Feature Costs**: There are two common ways to include cost: (1) additively, e.g. minusing the cost from the reward in RL [Li and Oliva 2021], and (2) to divide by the cost either in the reward [Kachuee et al. 2019] or in the CMI estimate [Gadgil et al. 2024]. We believe dividing by the cost is better. Intuitively, the adjusted scores represent the advantage of acquiring a feature per unit cost. And it enforces the desired property that features with zero cost are acquired. Additionally, like CMI, the scores and training of SEFA are independent of cost, scores are adjusted based on cost **after** calculation. So if feature costs vary over time, SEFA would not need to be retrained like RL methods that learn a policy.
>
> **Acquisition Budget**: Figure 3 shows the evaluation metrics throughout the acquisition - how the models would perform if the acquisition was ended after $n$ features. SEFA consistently performs best throughout the acquisition.
>
> # Theoretical Claims
> We have addressed this shared point in our response to Reviewer N1oA.
>
> # Hyperparameter Ranges
> **$\beta$**: The value of $\beta$ typically needs to be small, similar to a weight decay parameter. This is because it limits how much information from the features can be found in the latent space, papers such as Alemi et al. 2017 typically use values smaller than 0.01. In the sensitivity analysis in Figure 10 we see that if $\beta$ is too large, SEFA fails because it cannot predict effectively, and if $\beta$ is too low, then the latent space is not regularized enough. Based on this, we recommend 0.0001 - 0.01 as an effective range.
>
> **Number of Samples**: SEFA improves with the number of acquisition and training samples, shown by the sensitivity analyses in Figures 11 and 12. This is because during training it encourages a diverse latent space, and during acquisition the diversity can be fully sampled. We recommend 50-200 as a good tradeoff between acquisition performance and computation.
>
> **Number of Latent Components**: We use more than one latent component per feature so that there is a large capacity in the latent space for a rich representation of each feature. However, using too many increases the possibility of overfitting. Since each feature is only one value we recommend a value between 5-10. We tested 4, 6 and 8 as possible values and these were effective.
>
> **Other Hyperparameters**: The initial ranges were determined by manually finding good values on the synthetic datasets and using ranges around those. We started with broadly accepted values, e.g. 0.001 as the default PyTorch Adam learning rate.
>
> # References
> We agree with all suggestions and will add them. We will also cite Baehrens et al. 2009 and Simonyan et al. 2014 as gradient based feature attribution methods.
>
> # Feature Independence
> We have addressed this shared point in our response to Reviewer N1oA.
>
> # Variational Upper Bound
> The regularization term to minimize is $I(Z; X)$. This requires access to the marginal $p(z)=\int p(z|x)p(x)dx$, which is intractable. Instead we use the standard normal as a variational approximation since it produces an analytical upper bound. The derivation can be found in equations 13 and 14 of Alemi et al. 2017.
>
> # Noisy Features
> We have run Syn 1 (with two additional baselines) with added Gaussian noise with three standard deviations: 0.1, 0.2 and 0.4. The number of acquisitions to acquire the correct features are:
>
> |Model|No Noise|0.1|0.2|0.4|
> |---|---|---|---|---|
> |DIME|$4.079\pm0.057$|$4.688\pm0.211$|$4.703\pm0.276$|$5.368\pm0.267$|
> |EDDI|$9.183\pm0.187$|$8.988\pm0.285$|$9.216\pm0.136$|$9.382\pm0.245$|
> |Fixed MLP|$6.009\pm0.000$|$6.312\pm0.202$|$7.321\pm0.378$|$6.009\pm0.000$|
> |GDFS|$4.568\pm0.195$|$4.566\pm0.187$|$4.543\pm0.164$|$5.583\pm0.283$|
> |Opportunistic RL|$4.203\pm0.034$|$4.347\pm0.091$|$4.720\pm0.142$|$5.488\pm0.084$|
> |VAE|$6.593\pm0.085$|$6.866\pm0.041$|$7.005\pm0.086$|$7.095\pm0.078$|
> |GSMRL|$5.570\pm0.127$|$5.495\pm0.126$|$5.778\pm0.120$|$6.858\pm0.269$|
> |Random|$9.484\pm0.006$|$9.495\pm0.010$|$9.495\pm0.010$|$9.495\pm0.010$|
> |SEFA (ours)|$4.017\pm0.003$|$4.100\pm0.004$|$4.207\pm0.008$|$4.406\pm0.004$|
>
> The models get worse with more noise. SEFA remains the best model at all levels of noise. This is likely due to the latent space regularization removing noise between the feature space and latent space.

---

### Official Review · Reviewer_V2wP · 2025-03-19

**Overall Recommendation:** 4

**Summary:**

This paper considers active feature acquisition, which is a dynamic test-time selection (usually sequential) of features to make predictions on each test instances. The selected features are chosen independently for each considered test instance. The authors argue (with some theory) how prior methods based on conditional mutual information can fail in pathological cases, and how they might fail more generally. Instead the authors produce an information bottleneck inspired approach that utilizes a latent space to encode surrogate behavior for how unobserved variables might couple with selected features, in predicting the label. This relies on a decoupling of latent variables, one grouping for each candidate, along with a gradient-based acquisition function. There are extensive synthetic and real-world experiments to support the use of this method in practice.

**Claims And Evidence:**

The empirical results are strong, with sufficient comparisons against baselines and ablations, and results that seem significant over these baselines. I also really liked the explanation of why conditional mutual information might fail in proposition 4.1 and the surrounding discussion. This was well done.

**Essential References Not Discussed:**

See above. These alternate metrics instead of Shannon entropy are not in the critical path of the paper, but would strengthen the discussion

**Experimental Designs Or Analyses:**

Yes. There are a nice range of analyzes including both number of features needed to reach the optimal set (synthetic data) as well as accuracy vs number of acquired feature curves, and qualitative visualizations showing the "trajectories" of selected features.

**Methods And Evaluation Criteria:**

Yes

**Other Comments Or Suggestions:**

I like the visualizations showing the "trajectories" of the selected features. However, without the context of the scientific literature discussion in Section 6.3, I don't know how much Figure 4 is really adding. For someone without a medical background, the trajectories look random. This figure could be strengthened by annotating which features are actually scientifically important for each tumor location, so we can see that it is actually grabbing the important features, early on.

**Other Strengths And Weaknesses:**

- one additional baseline that I'm wondering about is, why didn't you compare against the baseline of selecting a *random* order of features, for each dataset?

I think Section 5 needs more exposition. It just gets right into the details of the design choices made, but this paper would be strengthened with additional discussion around each of these choices. In particular:
- What is the core intuition for z? Does it essentially serve as a surrogate for a marginalization over p(xU | xO), which would otherwise be too computationally expensive to consider?
- As discussed in the limitations, the factorization of the latent encoding is a huge assumption. This entirely decouples any relationship between the selected features. It is not clear to me at all why this is a good idea, intuitively. In fact, I'm a bit surprised that the empirical results are as good as they are, since the features do not seem to be coupled in any way. Am I missing something?
- The discussion around Shannon entropy set things up really nicely to argue how a different scoring metric is needed to better encourage maximal class probabilities on a single class. Yet, to me the choice of the gradient-based r(c,z,i) is still very much a heuristic. I think more discussion is needed around this scoring function. Why is this a good idea? Sensitivity of p(Y|z) is listed as the reason, but how does this in any way fulfill the property that Shannon entropy is lacking?

**Questions For Authors:**

Please see the comments above

## update after rebuttal
Thank you for your response. I maintain my score at accept

**Relation To Broader Scientific Literature:**

While I appreciate the discussion around Shannon entropy not necessarily encouraging a maximal probability most likely class, I think more discussion is needed here around alternate metrics. Sometimes Renyi entropies are used instead of Shannon entropy to encourage different properties. For instance the min-entropy exactly encourages maximum probability on a single class. Extrinsic Jensen-Shannon divergence is another example, as an alternative to mutual information that better distinguishes between likely hypotheses (see e.g., Naghshvar & Javidi 2012).

Naghshvar, M., & Javidi, T. (2012, July). Extrinsic Jensen-Shannon divergence with application in active hypothesis testing. In 2012 IEEE International Symposium on Information Theory Proceedings (pp. 2191-2195). IEEE.

**Theoretical Claims:**

I did not look carefully at the proofs

---

> ### Author Rebuttal · Authors · 2025-03-31
>
> Thank you for the positive review, we are grateful for the feedback, we answer your questions below. All changes below will be added.
>
> # Renyi Entropy
> Thank you for bringing this to our attention, we agree that Renyi entropies are relevant. In particular how the min-entropy relates to Shannon Entropy focussing on making low likelihoods lower (Section 4). We shall add these citations.
>
> That being said, Renyi entropies would still suffer from making myopic decisions. They would still fail on the Indicator for the same reasons as Shannon Entropy due to Proposition 4.1. Additionally it is highly non-trivial to incorporate these metrics into the SEFA architecture.
>
> # Random Selections
> We originally tested random orderings to determine whether datasets would benefit from a non-random ordering. Random orderings performed poorly so we did not present the results to save space. We have provided the original results in our response to Reviewer ABGZ in the Additional Baselines sections. The random ordering always performs worse than the Fixed Global MLP Ordering - the datasets benefit from non-random orders.
>
> # Using the Latent Space
> Marginalization over $p(\mathbf{x}\_{U} | \mathbf{x}\_{O})$ is possible, we do it in our Feature Space ablations (Table 1 and Table 3). Instead the choice to use the latent space was made to improve performance, due to the following reasons:
> - Acquisitions are not made using gradients calculated with feature level noise. By using latent space regularization we make acquisitions with a feature representation with noise removed containing only label relevant information.
> - We avoid the need to train a generative model, avoiding the complex feature distributions (e.g. multi-modal) with complex interdependencies. In latent space we can model the distribution as independent Gaussians and the predictor learns to predict using these, and handles the interdependencies. See more in the Feature Independence response for a further explanation.
> - Tabular data may contain features that are categorical (gradients are not meaningful), or features at different scales (the gradients are different scales), so gradients with respect to the raw features is not a reliable way to score them. We are able to train the encoders so that the latent components are all approximately the same scale (by regularizing the latent space) and real numbers, so that the gradients give reliable feature scores.
>
> In summary, by calculating scores using **representations** of the features, we avoid both the need to train a generative model, and the associated pitfalls (feature level noise, complex distributions, complex interdependencies, less meaningful gradients).
>
> # Feature Independence
> As suggested by the ICML Program Chairs we have addressed this shared point in our response to Reviewer N1oA. Briefly, the complex interdependencies are able to be accounted for by the predictor network. The encoders do not need to learn the couplings between the features, because the predictor is able to treat latent components **differently** based on observed features, and therefore the gradients change based on this.
>
> # Scoring Metric
> **Overcoming Shannon Entropy Drawbacks**: The two main properties that Shannon Entropy lacks are (1) considering unobserved values in the current decision and (2) focussing on identifying the most likely class. SEFA addresses the first property by taking an expectation of $r(c, \mathbf{z}, i)$ over $\mathbf{z}$ considering the possible latent realizations of the unobserved features. SEFA addresses the second property by using probability weighting (the outer sum over $p\_{\theta, \phi}(y| \mathbf{x}\_{O})$ in Equation 1).
>
> **Intuition behind  $r(c, \mathbf{z}, i)$**: As pointed out by Reviewer Pdip, a good perspective on $r(c, \mathbf{z}, i)$ is via Feature Attribution. Gradients are a common technique for explaining which features are most important for predicting a particular instance [Baehrens et al. 2009, Simonyan et al. 2014]. And LIME [Ribeiro et al. 2016] suggests attributions using locally linear approximations of the predicted probability for each class separately. So we take many samples from the latent space, representing possible future latent realizations. For each of these, $r(c, \mathbf{z}, i)$ uses gradients as local linear approximations, to calculate feature attributions via the latent components based on how important they are for predicting class $c$. After the mean over the latent samples is taken, we have feature attributions for each class, we take a sum over classes, weighted by the current predicted probability, $p\_{\theta, \phi}(y| \mathbf{x}\_{O})$, to give more focus to features that are relevant to more likely classes.
>
> # Figure 4
> Thank you for the suggestion, we agree that the figure can be improved to highlight the class relevant features. As suggested by the ICML Program Chairs, we have addressed this shared point in more detail in our response to Reviewer N1oA.

---

### Official Review · Reviewer_ABGZ · 2025-03-21

**Overall Recommendation:** 3

**Summary:**

The paper proposes a Stochastic Encodings for Feature Acquisition (SEFA) framework addressing the limitations of CMI and RL methods. SEFA uses an encoder-predictor architecture with intermediate stochastic latent variables. The architecture makes predictions and calculates the acquisition objective.

**Claims And Evidence:**

Yes.

**Essential References Not Discussed:**

NA

**Experimental Designs Or Analyses:**

Yes.

**Methods And Evaluation Criteria:**

Yes.

**Other Comments Or Suggestions:**

NA

**Other Strengths And Weaknesses:**

Strengths -

The theoretical proofs to establish the limitations of CMI and RL and the proposed SEFA framework, look solid. The performance look pretty good.

Weaknesses -

There's a high probability of encountering scalability issues with the proposed method and may turn out to be difficult to extend this to various other applications.

Some recent RL baselines can be used as baselines.

**Questions For Authors:**

Please check the weaknesses section.

**Relation To Broader Scientific Literature:**

The contributions are novel addressing the limitations of CMI and RL methods.

**Theoretical Claims:**

Yes.

---

> ### Author Rebuttal · Authors · 2025-03-31
>
> Thank you for the positive review, we are grateful for the feedback, we answer your questions below. All changes below will be added.
>
> # Scalability
> We provide the computational complexities for a single training step and single acquisition step for each model in Table 4 (Appendix H.3). The main takeaway is that SEFA scales better than half the methods at training time, better than the other half during acquisition and never the worst.
>
> As suggested by Reviewer Pdip, to add to this, we have timed each model to determine the wall-clock times for inference and training. The average times in seconds for a **single** acquisition are given below:
>
> |Model|Syn 1|Syn 2|Syn 3|
> |---|---|---|---|
> |DIME|$0.017\pm0.000$|$0.018\pm0.000$|$0.024\pm0.000$|
> |EDDI|$2.871\pm0.059$|$2.613\pm0.002$|$2.854\pm0.056$|
> |Fixed MLP|$0.014\pm0.000$|$0.011\pm0.000$|$0.017\pm0.000$|
> |GDFS|$0.018\pm0.000$|$0.015\pm0.001$|$0.022\pm0.000$|
> |Opportunistic RL|$0.029\pm0.001$|$0.021\pm0.001$|$0.032\pm0.000$|
> |VAE|$0.153\pm0.000$|$0.154\pm0.000$|$0.267\pm0.005$|
> |GSMRL|$0.047\pm0.000$|$0.040\pm0.000$|$0.055\pm0.000$|
> |Random|$0.014\pm0.000$|$0.011\pm0.000$|$0.018\pm0.001$|
> |SEFA (ours)|$0.246\pm0.001$|$0.165\pm0.001$|$0.240\pm0.002$|
>
> |Model|Cube|Bank Marketing|California Housing|MiniBooNE|
> |---|---|---|---|---|
> |DIME|$0.021\pm0.000$|$0.016\pm0.000$|$0.010\pm0.001$|$0.021\pm0.000$|
> |EDDI|$8.872\pm0.213$|$1.456\pm0.002$|$0.583\pm0.006$|$9.039\pm0.245$|
> |Fixed MLP|$0.020\pm0.000$|$0.012\pm0.000$|$0.006\pm0.000$|$0.019\pm0.000$|
> |GDFS|$0.021\pm0.000$|$0.017\pm0.002$|$0.008\pm0.000$|$0.020\pm0.000$|
> |Opportunistic RL|$0.031\pm0.000$|$0.022\pm0.001$|$0.010\pm0.000$|$0.034\pm0.001$|
> |VAE|$0.278\pm0.002$|$0.136\pm0.000$|$0.048\pm0.001$|$0.291\pm0.001$|
> |GSMRL|$0.061\pm0.000$|$0.035\pm0.001$|$0.021\pm0.000$|$0.045\pm0.000$|
> |Random|$0.019\pm0.000$|$0.012\pm0.000$|$0.006\pm0.000$|$0.017\pm0.000$|
> |SEFA (ours)|$0.861\pm0.004$|$0.130\pm0.001$|$0.077\pm0.000$|$0.314\pm0.002$|
> ||||||
> |Model|MNIST|Fashion MNIST|METABRIC|TCGA|
> |DIME|$0.024\pm0.000$|$0.023\pm0.000$|$0.003\pm0.000$|$0.010\pm0.000$|
> |EDDI|$12.360\pm0.029$|$8.543\pm0.200$|$0.108\pm0.001$|$0.765\pm0.011$|
> |Fixed MLP|$0.020\pm0.000$|$0.020\pm0.000$|$0.002\pm0.000$|$0.008\pm0.000$|
> |GDFS|$0.024\pm0.000$|$0.025\pm0.000$|$0.003\pm0.000$|$0.009\pm0.000$|
> |Opportunistic RL|$0.030\pm0.000$|$0.031\pm0.000$|$0.003\pm0.000$|$0.012\pm0.000$|
> |VAE|$0.452\pm0.007$|$0.452\pm0.007$|$0.009\pm0.000$|$0.031\pm0.001$|
> |GSMRL|$0.097\pm0.001$|$0.069\pm0.001$|$0.016\pm0.000$|$0.021\pm0.000$|
> |Random|$0.019\pm0.000$|$0.019\pm0.000$|$0.002\pm0.000$|$0.008\pm0.000$|
> |SEFA (ours)|$1.498\pm0.002$|$1.497\pm0.003$|$0.019\pm0.000$|$0.098\pm0.000$|
>
> As the number of classes increases (MiniBooNe to Cube and Cube to MNIST/Fashion MNIST), SEFA's acquisition time increases, despite the number of features staying the same, this matches Table 4. The time for Generative Models increases significantly with the number of features in line with Table 4. As expected, models with policy networks are fastest, and scale well with the number of features, but SEFA is never the slowest (faster than EDDI, often by a significant amount).
>
> The average time in milliseconds for a **single** training step (one gradient descent step) on Syn 1 is given below:
>
> |Model|Iteration Time (mS)|
> |---|---|
> |DIME|$35.156\pm0.081$|
> |EDDI|$7.290\pm0.028$|
> |Fixed MLP|$5.152\pm0.027$|
> |GDFS|$21.661\pm0.056$|
> |Opportunistic RL|$89.565\pm0.370$|
> |VAE|$6.761\pm0.075$|
> |GSMRL|$90.383\pm0.285$|
> |SEFA (ours)|$15.843\pm0.091$|
>
> Models that train by simulating acquisition are slowest to train despite being faster at inference time. Supervised models, including SEFA, are faster because they do not need to generate rollouts.
>
> # Additional Baselines
> Thank you for the suggestion, we have run the experiments on the following new baselines:
> - GSMRL [Li and Oliva 2021], a more recent and popular RL baseline for AFA.
> - A random ordering with an MLP predictor as requested by Reviewer V2wP (to show feature ordering is necessary).
>
> The results are given below:
>
> |Model|Synthetic 1|Synthetic 2|Synthetic 3|
> |---|---|---|---|
> |GSMRL|$5.570\pm0.127$|$6.227\pm0.185$|$8.199\pm0.067$|
> |Random|$9.484\pm0.006$|$9.499\pm0.005$|$9.987\pm0.008$|
> |SEFA (ours)|$4.017\pm0.003$|$4.098\pm0.007$|$5.081\pm0.021$|
>
> |Model|Cube|Bank Marketing|California Housing|MiniBooNE|
> |---|---|---|---|---|
> |GSMRL|$0.891\pm0.001$|$0.879\pm0.006$|$0.638\pm0.003$|$0.946\pm0.001$|
> |Random|$0.699\pm0.001$|$0.816\pm0.003$|$0.569\pm0.003$|$0.912\pm0.001$|
> |SEFA (ours)|$0.904\pm0.000$|$0.919\pm0.001$|$0.676\pm0.005$|$0.957\pm0.000$|
> ||||||
> |Model|MNIST|Fashion MNIST|METABRIC|TCGA|
> |GSMRL|$0.701\pm0.002$|$0.683\pm0.001$|$0.665\pm0.002$|$0.781\pm0.003$|
> |Random|$0.661\pm0.001$|$0.648\pm0.001$|$0.647\pm0.005$|$0.753\pm0.003$|
> |SEFA (ours)|$0.761\pm0.001$|$0.721\pm0.000$|$0.708\pm0.002$|$0.845\pm0.002$|
>
> SEFA is better than both new baselines. Random performs poorly in all cases.

---

### Decision · Program_Chairs · 2025-05-01

**Decision:**

Accept (poster)

**Comment:**

The reviewers thinks that this is a solid paper: it highlights the limitation of conditional mutual information-based methods, which motivated their novel approach of using latent space. Extensive synthetic and real-world experiments supported the use of this method in practice. Additional baselines, noisy feature experiments, as well as scalability evaluation in the rebuttal stage helped confirm the utility of the proposed methods.